# NO OUTLIER CHANNELS BUT WITH OUTLIER BLOCKS

**Shanwen Mao**[1,2]    **Hao Zhang**[1,2*]    **Jiasheng Li**[1,2]    **Haoyu Qiao**[1,2]
**Chenxin Cai**[1,2]    **Tingting Wu**[3]    **Jie Liu**[2,4]

[1]Harbin Institute of Technology, Harbin, China
[2]State Key Laboratory of Smart Farm Technologies and Systems, Harbin, China
[3]Harbin Engineering University, Harbin, China
[4]Harbin Institute of Technology (Shenzhen), Shenzhen, China

{24s103313, 2021113691, 24s103393, chxcai}@stu.hit.edu.cn
zhh1000@hit.edu.cn    ttwu@hrbeu.edu.cn    jieliu@hit.edu.cn

## ABSTRACT

With the rapid scaling of large language models, achieving efficient compression while maintaining model performance has become a critical challenge. To address the limitations of existing non-uniform quantization methods, which typically rely on fixed codebooks and require costly optimization, we propose a novel arbitrary bit-width non-uniform Quantization (NuBitQ). The framework enables flexible, layer-specific quantization strategies, significantly enhancing adaptability and efficiency. Notably, traditional outlier compensation methods used in uniform quantization are ill-suited for the anomalous distribution characteristics encountered in our context. To address this, we design a novel outlier evaluation metric that integrates weight perturbation, activation distribution, and perturbation propagation. Based on this metric, we further develop an Outlier Compensation Plugin (OCP) that implements multi-level, fine-grained outlier compensation strategies, effectively mitigating performance degradation caused by outliers. Our approach avoids direct complex Hessian computation and fine-tuning, offering strong applicability and scalability. Extensive experiments on multiple tasks and across various model series demonstrate the effectiveness of the proposed approach.[1]

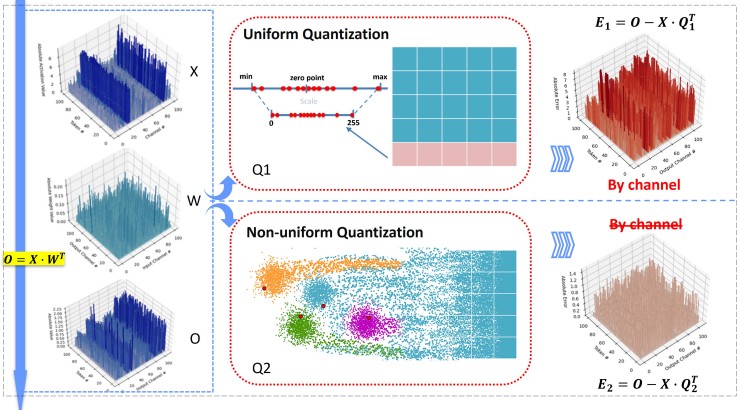

Figure 1: Comparison of uniform vs. non-uniform quantization effects. Left: original matrix multiplication; middle: uniform and non-uniform quantization; right: quantization error comparison.

---

*Corresponding author
[1]Code:https://github.com/maoshanwen/NuBitQ-OCP

# 1 INTRODUCTION

Large Language Models (LLMs) have achieved remarkable performance across various tasks. However, their deployment is significantly constrained by substantial demands on computation and memory. For instance, running LLaMA3-70B in FP16 precision requires at least four A100-40GB GPUs (Dubey et al., 2024), rendering real-world deployment costly and often impractical.

Quantization has emerged as a key technique to address the challenge by converting floating-point model parameters into low-precision integers, significantly reducing model size and inference latency with minimal performance loss. Uniform quantization, a predominant approach, divides the value range into equal intervals using per-channel scale and zero-point parameters (see the upper part of Figure 1). However, it is particularly vulnerable to outliers (rare but large-magnitude values), which lead to uneven quantization errors. This is especially problematic in channels with high activation variance, where these errors concentrate (illustrated as peaks in the "By channel" subplot at the upper right part of Figure 1), ultimately degrading model performance. To mitigate this, techniques such as LLM.int8 (Dettmers et al., 2022), AWQ (Lin et al., 2024b), and FlatQuant (Sun et al., 2025) introduce various outlier compensation mechanisms, including isolating outliers, adjusting precision for critical channels, and applying affine transformations.

In contrast, non-uniform quantization (lower part of Figure 1) leverages clustering-based methods to build more flexible codebooks that better match real-world distributions of model parameters. This significantly reduces quantization error (smaller and more uniform errors in the lower-right part of Figure 1). However, it introduces new challenges that error patterns become less predictable and more scattered across channels, resulting in novel outlier characteristics different from uniform quantization. Therefore, existing amplitude-based compensation strategies tailored for uniform quantization do not generalize well. Furthermore, state-of-the-art non-uniform quantization methods such as AQLM (Egiazarian et al., 2024) and VPTQ (Liu et al., 2024) typically rely on enlarged codebooks or residual fitting to reduce overall error, but often overlook error sensitivity. Although techniques like BCQ (Elangovan et al., 2025) and GPTVQ (Van Baalen et al., 2024) attempt to address this via layer-wise fine-tuning or Hessian-guided clustering, these approaches impose significant computational and memory overheads, making them less viable for large-scale models.

To overcome these limitations, we propose a quantization framework called NuBitQ, designed to support flexible, layer-wise quantization with high efficiency. Crucially, NuBitQ is complemented by a modular OCP that addresses the unique outlier distribution introduced by non-uniform quantization. We begin by formulating a layer-specific outlier impact metric that combines three dimensions: weight perturbation, activation distribution, and perturbation propagation. The metric enables precise identification of the layers containing critical outliers.

Building upon the insight, NuBitQ introduces a fine-grained, multi-codebook, multi-vector quantization strategy that adapts bit-width and codebook design per layer. In parallel, OCP implements a multi-granularity compensation scheme across (i) individual linear outliers, (ii) attention discrepancies in Transformer blocks, and (iii) global output distribution, together providing hierarchical and targeted error correction. Notably, our approach eliminates the need for layer-wise fine-tuning or Hessian matrix computation, making it scalable and practical for LLM compression.

Our main contributions are summarized as follows:

- We proposed NuBitQ, an efficient and flexible non-uniform quantization framework that supports arbitrary bit-widths and enables layer-wise differentiated strategies for improved adaptability and compression.

- We found the unique distribution of outliers in non-uniform quantization. We designed a plug-and-play OCP module based on a novel layer-wise outlier impact metric that jointly considers weight perturbation, activation statistics, and error propagation.

- We verified our method's effectiveness and scalability on representative LLMs, including LLaMA (Dubey et al., 2024), Qwen (Yang et al., 2025), and Gemma (Team et al., 2024). Results show near-lossless performance at 4-bit precision and strong competitiveness at lower bit-widths, outperforming existing non-uniform quantization methods.

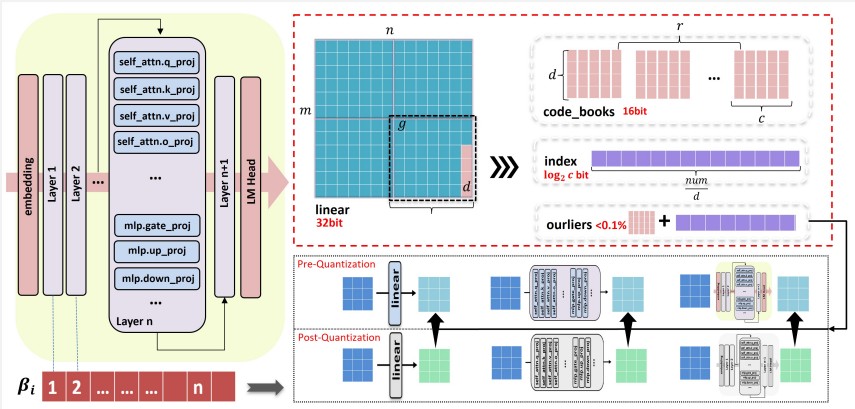

Figure 2: Overview of NuBitQ-OCP: The top-right part shows NuBitQ, a fine-grained layer-wise quantization method. The bottom-right and left parts together constitute OCP: the bottom-right illustrates compensation methods at different granularities, while the left side displays outlier scores that highlight outliers to guide the compensation.

## 2 RELATED WORK

### 2.1 NON-UNIFORM QUANTIZATION

Non-uniform quantization uses variable intervals based on data distribution, reducing global error and enabling lower-bit quantization. For instance, Tseng et al. (2024) applies vector quantization to exploit the spherical sub-Gaussian distribution of incoherent weights, achieving higher precision but relying on a fixed codebook. VPTQ (Liu et al., 2024) enhances granularity with channel-independent second-order optimization, yet it uses only two codebooks for original values and residuals. AQLM (Egiazarian et al., 2024) advances LLM compression through additive quantization and layer-wise fine-tuning while depending on fixed codebook sizes per layer. GPTVQ (Van Baalen et al., 2024) improves quantization for LLMs by increasing dimensions and incorporating per-layer output MSE and Hessian information, though Hessian computations incur significant costs. In contrast, our method avoids Hessian computations and, through adaptive codebook arrangements, achieves flexible bit-width compression, yielding optimal performance at 4-bit results.

### 2.2 OUTLIERS

Research shows some feature dimensions in LLM have outlier values much larger than other channels (Kovaleva et al., 2021), making quantization challenging. Mitigation methods mainly fall into two types: one relies on first-order statistics of weight and activation values before and after quantization, such as magnitude, to identify outlier channels (Dettmers et al., 2023)or uses smoothing factors to reduce quantization errors (Xiao et al., 2023). The other employs second-order statistics, utilizing Hessian matrices or their variants, such as Fisher Information Matrices (Kim et al., 2024), to reduce outlier impact through Hessian weighting (Frantar et al., 2023), rotation transformations (Ma et al., 2024) (Lin et al., 2024a), or other affine transformations (Huang et al., 2024) (Saxena et al., 2025). These methods target outlier channels. Studies (Gong et al., 2024) show that outlier handling is heavily influenced by the quantization method, with clipping in uniform quantization causing prominent outlier channels, while non-uniform quantization produces more dispersed and smaller errors that cannot be simply identified by channels. Dedicated compensation methods for outliers in non-uniform quantization remain limited. We proposes a specialized optimization strategy to better handle outlier blocks in non-uniform quantization, enabling effective compression below 4 bits.

## 3 METHODOLOGY

### 3.1 NUBITQ: FLEXIBLE QUANTIZATION

To address the challenges outlined in the previous section, we designed NuBitQ, a flexible non-uniform quantization that supports layer-wise customization and ultra-low bit-widths. NuBitQ builds upon the *codebook + index* paradigm, with several key improvements aimed at achieving a better trade-off between compression ratio and model accuracy.

As illustrated in Figure 2, given a linear layer weight matrix of size $n \times m$, we first divide it into $\frac{n \times m}{d}$ subvectors of dimension $d$. These subvectors evenly assigned to $g$ groups, each equipped with a learnable scaling factor $q$. For each group, we construct a codebook containing $c$ cluster centers, each of which is a vector of length $d$.

To improve clustering quality, we apply beam search with width $b$ during the $k$-means procedure. Furthermore, we introduce a residual quantization strategy by using $r$ sequential codebooks: the first codebook encodes the original subvector, while each subsequent codebook encodes the residual error from the previous quantization step. As a result, each subvector is approximated by a sequence of $r$ indices, each pointing to a codeword in its corresponding codebook. Hence, the compressed model stores $r \times c \times d \times g$ codebook parameters and $r \times \frac{n \times m}{d}$ indices, replacing the original floating-point weights.

In terms of memory usage, the original weight matrix, stored as 32-bit floats, requires $n \times m \times 32$ bits. In the quantized form, each index requires $\log_2 c$ bits, resulting in a total of $r \times \frac{n \times m}{d} \times \log_2 c$ bits. The codebooks, stored at 16-bit precision, consume $r \times c \times d \times g \times 16$ bits in total. Notably, when the weight matrix is large, the codebook memory becomes negligible compared to the index storage. Therefore, the total quantized memory can be approximated as the sum of index and codebook storage. We denote the compression ratio $R$ as the ratio of quantized memory to original memory: $R = \frac{r \times \frac{\log_2 c}{d} + \frac{r \times c \times d \times g \times 16}{n \times m}}{32}$. When $n \times m$ is sufficiently large, the second term becomes negligible, and we further reduce memory by re-clustering the codebooks of each group. The compression ratio simplifies to

$$R \approx \frac{r \times \log_2 c}{32 \times d}. \tag{1}$$

By performing a grid search over the number of codebooks $r$, the number of cluster centers $c$, and the subvector dimension $d$, it allows for flexible adjustment of the compression-performance trade-off, supporting diverse quantization configurations from high-precision to ultra-low-bit settings.

### 3.2 OCP: OUTLIER COMPENSATION PLUG-IN

#### 3.2.1 OUTLIER PATTERN

In low-bit uniform quantization, large errors often concentrate in specific channels, forming clear outlier channels that significantly degrade model accuracy. However, in non-uniform quantization, errors tend to be generally smaller and more dispersed, making standard outlier detection methods less effective. Here, we define *outliers* as quantization errors that substantially impact accuracy. A common heuristic is to use the difference between quantized and original weights to identify potential outliers. Yet, further analysis reveals these errors are not equally harmful—some contribute to performance degradation, exhibiting patterns clustered at the block level, while others are relatively benign.

To investigate this, we analyze the impact of 2-bit quantization on LLaMA3-8B's Transformer blocks 0 through 31. We observe significant variability in quantization sensitivity across blocks: for example, independently quantizing Transformer block 1 causes the largest increase in perplexity (PPL), whereas other blocks show minor effects (see Figure 3a). Simultaneously, we provide a detailed breakdown of the internal blocks (see Figure 3b) and perform experiments with different input data (see Figure 3c). This suggests that outliers do not appear as isolated channels but rather as localized blocks.

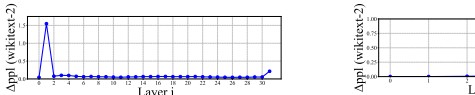

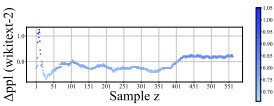

(a) Quantize Only Layer i to 2 bits     (b) Only Layer1.linearj to 2 bits     (c) Layer1.linear6 2bits on Sample z

Figure 3: From observing the outlier Transformer blocks of the entire model, to examining the outlier sublinear within outlier Transformer block, and finally to analyzing the outlier sample inputs of the outlier sublinear.

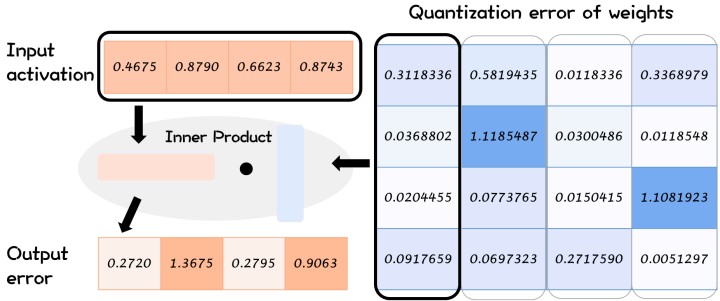

Figure 4: Simplified Diagram of Output Error Caused by Non-Uniform Quantization

To illustrate how non-uniform quantization errors develop into outliers at the block level, consider the following simplified example. Suppose the vector dimension is $d = 4$, and after quantization with codebook replacement, the quantization errors for four vectors in a block are shown in Figure 4.

Although the overall block error is low due to vector clustering, some individual vectors incur significant single-point deviations to fit their assigned codebook entries, leading to amplified output errors. These scattered errors can be further magnified by input activations and propagated through subsequent layers, eventually developing into significant outliers that markedly affect model performance.

Overall, the impact of errors between quantized and original weights on model performance varies. The outliers that cause performance drops are unevenly spread across blocks and depend on the input data.

### 3.2.2 OUTLIER SCORE

From the above observations, we develop a theoretical metric to quantify the impact of weight outliers within each Transformer block. The metric accounts for hierarchical structure and input-output correlations to better capture the propagation of quantization-induced errors.

Consider the $i$-th Transformer block containing 7 linear sublayers. We define the original and quantized weights for the $j$-th sublayer as $W_{i,j}^\star$ and $Q(W_{i,j})$, respectively. The difference between them is $\Delta W_{i,j} = W_{i,j}^\star - Q(W_{i,j})$, regarding as the weight perturbation. The perturbation propagates through the layer's nonlinear computations, and ultimately influences the model output. We describe its propagation via Jacobian matrices (Novak et al., 2018):

$$\Delta Y_L = J_{i\to L} \sum_{j=1}^{7} J_{i,j}(\Delta W_{i,j}), \tag{2}$$

where $J_{i,j}$ represents the local sensitivity of the $j$-th sublayer's weight perturbation to the layer output, and $J_{i\to L}$ denotes the aggregated sensitivity from layer $i$ to the final model output. To simplify computation, we approximate the outlier impact using the expected squared Frobenius norm (Sato & Suzuki, 2024) of the output perturbation:

$$I_i := \mathbb{E}\|\Delta Y_L\|_F^2 \approx \sum_{j=1}^{7} \mathbb{E}\left[(\Delta W_{i,j})^\top M_{i,j}(\Delta W_{i,j})\right], \tag{3}$$

where $M_{i,j} = \left[ J_{i,j}^\top J_{i \to L}^\top J_{i \to L} J_{i,j} \right]$ represents both the sensitivity of the sublayer to perturbations and their amplification through subsequent layers. Specifically, the sublayer's sensitivity is mainly determined by the input activation covariance matrix $C_{i,j} := \mathbb{E} \left[ x_{i,j} x_{i,j}^\top \right]$, which reflects how different input directions affect the output locally. The amplification of perturbations as they propagate to the final output is mostly governed by the norms of downstream weights, approximated by the product of squared Frobenius norms $\prod_{k=i+1}^{L} \| W_k \|_F^2$.

Following Hessian trace approximation techniques Dong et al. (2020), we decompose the Eq. (3) into three interpretable components:

- The perturbation magnitude: $\| \Delta W_{i,j} \|_F^2$.
- The trace of the input activation covariance: $\mathrm{tr}(C_{i,j})$, representing sensitivity to weight perturbations.
- The product of squared Frobenius norms of subsequent layers' weights: $\prod_{k=i+1}^{L} \| W_k \|_F^2$, describing hierarchical amplification.

To better integrate values of different magnitudes and mitigate the impact of numerical disparities, we adopt the logarithmic form and define the logarithmic *outlier score* for Transformer block $i$ as

$$\beta_i = \sum_{j=1}^{7} \left( \log \| \Delta W_{i,j} \|_F^2 + \log \mathrm{tr}(C_{i,j}) + \sum_{k=i+1}^{L} \log \| W_k \|_F^2 \right) \tag{4}$$

The metric integrates perturbation magnitude, input statistics, and inter-layer propagation effects, effectively reflecting the potential influence of outliers in each Transformer block on overall model performance. More detailed analysis of our proposed outlier score is in Appendix D.

### 3.2.3 OUTLIER COMPENSATION

Based on the guidance of outlier scoring, we maintain an outlier codebook pool and employ a sliding window mechanism to select codebook entries that align with our compensation strategies. We design three hierarchical compensation strategies to mitigate performance degradation caused by quantization. These methods target different structural levels based on the severity of outlier impact and aim to restore model performance with minimal overhead. Each strategy optimizes a compensation term, either a codebook or weight delta, so that the compensated quantized weight $Q(W) + \Delta W$ better approximates the original weight $w^\star$.

*MSE Minimization (Linear-sublayer level).* For the $i$-th Transformer layer's $j$-th linear sublayer, given input activation $x_{i,j} \in \mathbb{R}^{n \times h}$ (batch size $n$, feature dimension $h$), we optimize the compensation perturbation $\Delta W_{i,j}$ by minimizing the expected output mean squared error induced by quantization perturbations, where we let $Q^\star = Q(W_{i,j}) + \Delta W_{i,j}$ denote the combined quantized weight and compensation perturbation. Then the optimization problem is

$$\Delta W_{i,j}^\star = \arg \min_{\Delta W_{i,j}} \mathbb{E}_{x_{i,j}} \left\| x_{i,j} W_{i,j}^\star - x_{i,j} Q^\star \right\|_F^2. \tag{5}$$

The strategy leverages input activation statistics to finely calibrate weight compensation, suitable for sublayers with prominent outlier scores.

*Attention Score Deviation Minimization (Transformer-block level).* For the $i$-th Transformer blocks with self-attention, we minimize the difference in attention score matrices before and after compensation:

$$\theta_i^\star = \arg \min_{\theta_i} \| A_i^\star - A_i(\theta_i) \|_F^2, \tag{6}$$

where $A_i(\theta_i)$ denotes the attention score matrix after compensation. The strategy preserves the model's self-attention capability and works well for structurally complex layers with concentrated, stable outlier perturbations.

*KL Divergence Minimization (Whole-model level).* For coarse-grained global compensation, we minimize the KL divergence between the original and quantized model output distributions:

$$\theta^\star = \arg \min_{\theta} \mathbb{E}_{x_{\leq t}} \left[ D_{\mathrm{KL}} \left( p^\star(\cdot \,|\, x_{\leq t}), p(\cdot \,|\, x_{\leq t}; \theta) \right) \right], \tag{7}$$

Table 1: Perplexity results of various bit quantization methods applied to multiple language models. Quantization method used (A = AQLM, V = VPTQ, G = GPTVQ, N = ours, O=OCP).

| #Bits | | Llama3-8B | | Qwen3-8B | | Gemma2-9B | | Qwen3-14B | | Gemma2-27B | | Llama3-70B | |
|---|---|---|---|---|---|---|---|---|---|---|---|---|---|
| | | Wiki ↓ | Ptb ↓ | Wiki ↓ | Ptb ↓ | Wiki ↓ | Ptb ↓ | Wiki ↓ | Ptb ↓ | Wiki ↓ | Ptb ↓ | Wiki ↓ | Ptb ↓ |
| 16 | - | 5.57 | 8.92 | 8.58 | 13.71 | 10.69 | 37.03 | 7.58 | 12.09 | 5.39 | 17.49 | 2.53 | 7.06 |
| 4 | A | 6.04 | 9.50 | 8.91 | 14.59 | 10.91 | 39.41 | 8.06 | 13.68 | 6.02 | 19.63 | 2.85 | 7.68 |
| | V | 6.10 | 9.60 | 22.67 | 35.11 | 66.02 | 274.77 | 16.20 | 27.99 | 54.73 | 133.90 | 3.12 | 7.85 |
| | G | 5.81 | 9.18 | 8.86 | 14.21 | 10.70 | 37.35 | 8.01 | 14.48 | 5.91 | 18.28 | 2.63 | 7.61 |
| | **N** | **5.79** | **9.14** | **8.81** | **14.17** | **10.68** | **37.39** | **7.86** | **12.90** | **5.69** | **18.06** | **2.59** | **7.54** |
| 3 | A | 7.05 | 11.11 | 9.49 | 16.63 | 11.43 | 42.14 | 8.91 | 14.14 | 6.73 | 21.40 | 3.45 | 8.11 |
| | A+O | 6.93 | 10.99 | 9.33 | 16.33 | 11.32 | 42.07 | 8.79 | 14.02 | 6.68 | 21.37 | 3.42 | 8.09 |
| | V | 7.01 | 10.46 | 30.43 | 49.92 | 185.33 | 939.84 | 23.55 | 37.10 | 113.40 | 441.72 | 3.62 | 9.28 |
| | V+O | 6.86 | 10.23 | 12.34 | 25.23 | 14.44 | 53.06 | 19.49 | 19.32 | 8.91 | 46.98 | 3.59 | 9.27 |
| | G | 6.03 | 9.98 | 9.42 | 15.14 | 11.13 | 39.66 | 8.99 | 14.46 | 6.65 | 22.17 | 3.07 | 7.87 |
| | G+O | 6.12 | 9.73 | 9.33 | 14.99 | 11.04 | 38.73 | 8.85 | 14.35 | 6.58 | 21.88 | 3.05 | 7.83 |
| | **N+O** | **5.66** | **8.98** | **8.87** | **13.79** | **10.80** | **38.09** | **8.63** | **13.27** | **6.22** | **20.54** | **2.98** | **7.60** |
| 2 | A | 7.28 | 11.61 | 10.15 | 17.54 | 12.27 | 49.43 | 9.75 | 15.59 | 7.97 | 27.95 | 5.52 | 8.59 |
| | A+O | 7.07 | 11.40 | 9.85 | 17.03 | 12.04 | 49.28 | 9.52 | 15.44 | 7.85 | 26.76 | 5.44 | 8.51 |
| | V | 9.19 | 12.77 | 1.65e6 | 8.08e6 | 3.27e6 | 4.93e6 | 1.08e6 | 2.22e6 | 1.28e6 | 3.79e6 | 6.19 | 10.75 |
| | V+O | 8.92 | 12.02 | 5.34e4 | 3.64e4 | 1.06e5 | 8.83e5 | 8.98e5 | 9.55e5 | 7.65e5 | 6.93e5 | 6.13 | 10.67 |
| | G | 10.34 | 21.89 | 13.19 | 28.07 | 13.83 | 56.30 | 9.83 | 22.52 | 7.47 | 27.24 | 5.43 | 8.48 |
| | G+O | 9.14 | 20.69 | 12.91 | 27.69 | 13.41 | 54.85 | 9.61 | 22.29 | 7.13 | 27.02 | 5.32 | 8.39 |
| | **N+O** | **6.42** | **9.74** | **9.35** | **15.32** | **11.45** | **47.98** | **9.33** | **14.99** | **6.94** | **26.74** | **4.99** | **8.03** |

where $p^\star$ represents the output distribution of the original model, $p$ denotes the output distribution of the quantized model, and $D_{KL}$ is the Kullback-Leibler divergence, used to measure the difference between the two probability distributions. The strategy directly maximizes the probability of generating correct tokens, suitable for overall performance restoration and preserving high-level semantic consistency.

These strategies adjust compensation from fine to coarse granularity based on outlier scores and resources, mapping the granularity differences as shown in the Outlier Pattern. Moreover, the selected data samples are taken from the few outlier samples identified above. Our main goal is to reduce overall errors while increasing the chance of generating correct tokens. The improvement comes from the optimization goals, not the specific compensation methods. Follow-up experiments show all compensation methods perform better than traditional quantization.

## 4 EXPERIMENTS

### 4.1 EXPERIMENTAL SETUP

We follow the evaluation protocol of LLMCBench (Yang et al., 2024) to compare the performance of our method against AQLM (Egiazarian et al., 2024), VPTQ (Liu et al., 2024), and GPTVQ (Van Baalen et al., 2024) in terms of knowledge ability, reasoning ability, and reliability. The evaluation tasks include perplexity measurement on WikiText2 (Merity et al., 2017) and Penn Treebank (PTB) (Prasad et al., 2014), as well as multiple benchmarks such as MMLU (Hendrycks et al., 2021), QNLI (Rajpurkar et al., 2016), MNLI (Williams et al., 2018), AdvGLUE (Wang et al., 2021), and TruthfulQA (Lin et al., 2022). Detailed experimental settings are in Appendix E.1.

### 4.2 MAIN RESULTS

#### 4.2.1 4-BIT QUANTIZATION COMPARISON

Table 1(4-bit) shows that, at 4-bit without using the OCP, our method achieves the lowest perplexity among all quantization methods. This indicates that NuBitQ effectively preserves the model's generative and comprehension capabilities while compressing it, highlighting its superiority. As the model size increases, the perplexity of our method approaches that of FP16. This suggests that our approach adapts better to complex language patterns in larger models and retains important feature information more effectively during quantization, thereby reducing performance loss. It should

Table 2: Accuracy of different bit quantization methods on Llama3-8B across various tasks.

| Method | #Bits | Knowledge (%) ↑ | | | | | Inference (%) ↑ | | Trustworthiness (%) ↑ | | |
|---|---|---|---|---|---|---|---|---|---|---|---|
| | | Hums. | STEM | Social | Other | Avg. | QNLI | MNLI | advglu | T.mc1 | T.mc2 |
| - | 16 | 55.05 | 53.38 | 73.90 | 69.59 | 62.18 | 50.90 | 40.95 | 44.17 | 28.52 | 46.75 |
| AQLM | 4 | 49.34 | 51.90 | 67.60 | 65.05 | 57.83 | 49.81 | 36.68 | 42.95 | 25.46 | 44.37 |
| VPTQ | 4 | 52.77 | 50.83 | 70.69 | 66.04 | 59.34 | 50.84 | 36.18 | 43.50 | 27.30 | 45.87 |
| GPTVQ | 4 | 51.80 | 48.87 | 69.39 | 65.55 | 58.20 | 51.05 | 39.09 | 44.58 | 25.46 | 42.93 |
| **NuBitQ** | **4** | **53.43** | **52.82** | **71.99** | **68.66** | **60.88** | **50.73** | **42.05** | **44.04** | **25.95** | **44.83** |
| AQLM | 3 | 29.59 | 29.82 | 35.53 | 31.83 | 31.41 | 49.32 | 34.56 | 43.36 | 25.21 | 43.29 |
| AQLM+OCP | 3 | 29.54 | 30.15 | 35.72 | 31.92 | 31.58 | 49.27 | 34.67 | 43.22 | 25.34 | 43.32 |
| VPTQ | 3 | 50.75 | 47.78 | 67.01 | 63.82 | 56.69 | 50.58 | 38.45 | 43.26 | 25.83 | 43.15 |
| VPTQ+OCP | 3 | 50.83 | 49.11 | 66.92 | 63.93 | 57.70 | 50.12 | 38.97 | 43.76 | 26.33 | 44.18 |
| GPTVQ | 3 | 43.68 | 42.35 | 58.01 | 55.06 | 50.71 | 50.71 | 35.06 | 44.58 | 28.03 | 44.96 |
| GPTVQ+OCP | 3 | 43.49 | 41.68 | 57.59 | 54.84 | 48.81 | 50.57 | 33.74 | 43.50 | 27.78 | 44.97 |
| **NuBitQ+OCP** | **3** | **54.75** | **53.02** | **74.16** | **69.65** | **62.07** | **50.71** | **40.79** | **44.72** | **27.17** | **46.15** |
| AQLM | 2 | 42.23 | 39.63 | 50.02 | 47.84 | 44.67 | 49.75 | 36.41 | 43.36 | 25.58 | 42.40 |
| AQLM+OCP | 2 | 40.09 | 41.83 | 51.51 | 50.74 | 45.63 | 49.62 | 38.90 | 43.36 | 25.34 | 42.85 |
| VPTQ | 2 | 42.12 | 38.84 | 50.99 | 42.92 | 43.69 | 49.21 | 34.54 | 43.22 | 25.21 | 46.35 |
| VPTQ+OCP | 2 | 43.18 | 40.92 | 52.17 | 45.86 | 45.53 | 49.62 | 36.78 | 42.94 | 26.89 | 46.90 |
| GPTVQ | 2 | 28.86 | 24.12 | 30.94 | 24.89 | 26.81 | 48.49 | 33.67 | 43.36 | 26.56 | 47.79 |
| GPTVQ+OCP | 2 | 24.10 | 28.96 | 30.78 | 24.86 | 26.78 | 48.12 | 33.46 | 43.36 | 26.19 | 47.80 |
| **NuBitQ+OCP** | **2** | **49.48** | **48.61** | **67.47** | **64.81** | **56.77** | **49.66** | **49.60** | **43.77** | **29.63** | **47.58** |

be clarified that, since the VPTQ method does not provide the Hessian matrix data required by its own approach for Qwen3 and Gemma 2, the results obtained without Hessian tuning lead to VPTQ quantization performance being lower than the theoretical values.

Table 2 shows that NuBitQ achieves the highest accuracy at 4-bit. This indicates that not only does our method excel in perplexity, but it also demonstrates competitive accuracy in task performance, enabling effective quantization. Notably, our method's accuracy at the underlined positions exceeds that of FP16, which is a pleasant surprise post-quantization. The result further supports the effectiveness of our approach and showcases the potential of our quantization strategy in specific scenarios.

### 4.2.2 ULTRA-LOW-BIT QUANTIZATION COMPARISON

We integrate our OCP with various methods for quantization below 4 bits across different models.The perplexity results are presented in Table 1. The results indicate that our method achieves the lowest perplexity. Additionally, when other methods integrate our OCP, the perplexity can be further reduced.However, since AQLM and GPTVQ themselves employ fine-tuning techniques for optimization, the scope for improvement with OCP is limited. For VPTQ on Qwen3-8B and Gemma2-9B, as its own optimization was not enabled, our compensation method significantly reduced the perplexity. Table 2 reports the accuracy results after quantizing the Llama3-8B model. Our method combined with OCP achieves the highest accuracy, while other methods using our OCP occasionally show a decrease in accuracy on certain tasks; however, the overall accuracy remains improved.

From these experiments, we observe that the outlier-compensated non-uniform quantization methods consistently outperform their original counterparts, particularly when the model size is smaller (such as 7B and 13B) or when the bit-width is lower (such as 2-bit). The phenomenon indicates that OCP serves as a powerful enhancement for non-uniform quantization.

### 4.3 ABLATION STUDIES

### 4.3.1 IMPACT OF QUANTIZATION PARAMETERS

In the 4-bit quantization setting of the LLaMA3-8B model, we quantized only the 7th linear layer of the model's 0-th layer and observed the changes in perplexity on the WikiText2 dataset, as shown in Figure 5a. Intuitively, larger values of parameters $r$, $c$, and $g$ are preferable, while a smaller value of parameter $d$ is advantageous. However, under the constraint of maintaining a 4-bit quantization compression ratio, we can see that: it clearly shows a significant impact of the $r$ parameter on perplexity, with an optimal range identified. It indicates that smaller values of d yield better performance, but

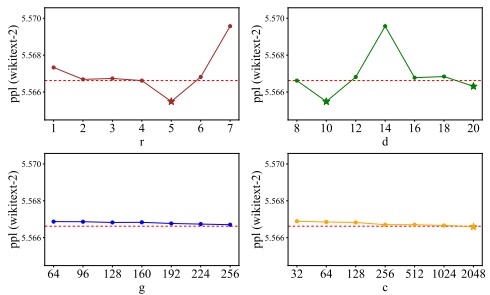

(a) Effects of Hyperparameter Selection quantization.

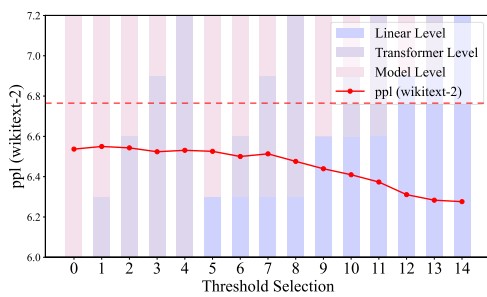

(b) Different combinations of thresholds for the 2-bit NuBitQ quantized LLaMA3-8B are presented.

Figure 5: The red dashed line on the left indicates the baseline perplexity before quantization, and on the right indicates the perplexity before compensation after quantization. The bars indicate the proportions of three compensation strategies utilized across a total of 32 layers.

Table 3: Overall ablation study on compensation methods and clustering configurations.

(a) Compensation strategies on the 31st layer of LLaMA3-8B

(b) Computation time (T) and memory usage (M) for clustering vector lengths in the 7th linear layer

| Strategy | Time (s) | Mem (%) | $\Delta$PPL $\uparrow$ |  |  | FP16 | d=4 | d=8 | d=16 | d=32 | d=64 |
|---|---|---|---|---|---|---|---|---|---|---|---|
| Random | 7.43 | 1.00% | 1.00× | Ada | T(ms) | 134.3 | 63.6 | 63.7 | 91.3 | 93.5 | 98.8 |
| Linear | 51.65 | 1.00% | 5.27× |  | M(MB) | 378 | 144 | 144 | 145 | 148 | 176 |
| Transformer | 15.53 | 0.29% | 2.26× | Ampere | T(ms) | 84.0 | 50.1 | 53.0 | 54.9 | 56.5 | 63.0 |
| Model | 8.33 | 0.14% | 2.12× |  | M(MB) | 408 | 144 | 144 | 144 | 148 | 176 |

both excessively small or large values can lead to increased perplexity. The effects of parameters $g$ and $c$ on model perplexity are relatively minor, resulting in flatter curves. This suggests that the $g$ and $c$ parameters have a smaller and more stable impact on quantization performance, while the $r$ and $d$ parameters play a dominant role in overall performance.

It is worth noting that points marked with pentagons in the figure indicate that the post-quantization perplexity is lower than the baseline before quantization, which is somewhat counterintuitive. Subsequently, we attempted to increase the $g$ value in the parameter combination and found that the quantization performance actually declined (in Appendix E.3.3), indicating that such results are somewhat coincidental. However, this also highlights the significant potential of non-uniform quantization parameter combinations in performance optimization.

### 4.3.2 IMPACT OF COMPENSATION STRATEGY

Table 3a illustrates the impact of different compensation strategies. Using the random compensation strategy as a baseline, which compensates for 1% of the original memory. The results indicate that the linear-level compensation consumes the most memory and time, yet it provides a compensation effect that is 5.27 times greater than that of the random strategy. In contrast, the model-level compensation uses the least memory, occupying only 0.14%, while achieving a performance improvement of 2.12 times over the random strategy.

Specifically, linear compensation requires more resources but provides finer correction; model compensation uses less memory but is coarser; Transformer compensation balances both and suits general layers. Choose compensation based on needs to improve performance and save resources.

### 4.3.3 IMPACT OF COMPENSATION THRESHOLD

Different combinations of thresholds lead to varying proportions of compensation strategies. A lower $\theta_1$ indicates a greater use of model-level compensation, while a lower $\theta_2$ signifies a greater use of linear-level compensation.

Figure 5b presents our results: when the threshold combinations is set to 0, we achieve a significant reduction in perplexity. As the threshold combinations increase from 1-4, incorporating more Transformer-level compensation, the perplexity shows an overall decreasing trend, although at a slower pace. When the threshold combinations range from 8-14, compensating for more linear-level layers, the perplexity decreases more rapidly compared to the previous range.

These results highlight the importance of selecting appropriate thresholds for optimizing model performance and reducing perplexity. If memory is a constraint, we can opt for a lower threshold $\theta_1$, while if accuracy is paramount, a lower threshold $\theta_2$ is more appropriate.

### 4.3.4 IMPACT OF VECTOR LENGTH SELECTION

Using FP16 precision as a baseline, we compared computation times and peak memory usage for various vector length settings ($d = 4, 8, 16, 32, 64$) across two hardware architectures: Ada Lovelace and Ampere. The results are shown in Table 3b. For hyperparameter selection, to maintain a 2-bit compression rate, we fix $r = 2$ and let $c$ change with the vector length $d$. This design ensures that the compression effect is preserved while adapting to different vector length settings.

The findings indicate that under the Ada Lovelace architecture, using vector lengths of 4 or 8 results in computation speeds approximately 2.1 times faster than FP16, while memory consumption decreases by about one-third. As the vector length increases, the speed decreases and memory usage increases, illustrating a trade-off between performance and resource utilization. A similar trend is observed in the Ampere architecture. Overall, shorter vector lengths (4 or 8) contribute to enhanced inference efficiency and reduced memory consumption, suitable for efficient deployment.

## 5 CONCLUSION

In this paper, we introduced a novel non-uniform quantization framework NuBitQ. The framework enables flexible layer-specific quantization strategies, outperforming existing methods at 4-bit precision. To enhance the effectiveness of ultra-low-bit quantization, we developed a modular OCP that leverages layer-specific metrics to maintain near-lossless performance. Experiments demonstrate that the NuBitQ-OCP framework excels in various natural language tasks. Currently, our research is focused on LLMs. In future we could explore multimodal models to improve scalability and effectiveness in more complex scenarios.

## 6 ACKNOWLEDGEMENTS

This work is supported in part by the Project of Laboratory of Advanced Agricultural Sciences of Heilongjiang Province under Grant No. ZY04JD05-010, the National Natural Science Foundation of China under Grant No. 62350710797.

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

## A DIFFERENCES IN METHODS

In this section, we examine the differences among several quantization methods. For brevity, we refer to these methods in Table 4 using their respective abbreviations: QUIP (Tseng et al., 2024), VPTQ (Liu et al., 2024), AQLM (Egiazarian et al., 2024), GPTVQ (Van Baalen et al., 2024), PCDVQ (Yue et al., 2025a), Rotation (Fifty et al., 2025), SSVQ (Yue et al., 2025b), SpinQuant (Liu et al., 2025), OSTQuant (Hu et al., 2025), and CBQ (Ding et al., 2025).

Since we are focusing on non-uniform quantization for large language models, our experimental comparisons select several baselines from the latest non-uniform quantization methods. Among them, PSSVQ and Rotation are intended for visual models, and PCDVQ has not publicly released its code.

**Notes:**

[1] Uses two codebooks; channel-independent 2nd-order optimization.

[2] Additive quantization; layer-wise fine-tuning.

Table 4: Comparison of Quantization Methods on Key Features. Non-uniform Q: whether the quantization is non-uniform; Layer Q: layer-wise quantization; Channel Q: channel-wise quantization; Outliers: handling of outlier values; Adaptive CB: use of adaptive codebook; Any BW: support for any bit-width; 2nd-order Info: use of second-order information in optimization.

| Method | Non-uniform Q | Layer Q | Channel Q | Outliers | Adaptive CB | Any BW | 2nd-order Info | Notes |
|---|---|---|---|---|---|---|---|---|
| VPTQ | ✓ | ✗ | ✗ | Partial | ✗ | ✗ | ✓ | [1] |
| AQLM | ✓ | ✓ | ✗ | ✗ | ✗ | ✗ | ✗ | [2] |
| GPTVQ | ✓ | ✓ | ✗ | ✗ | ✗ | ✗ | ✓ | [3] |
| PCDVQ | ✓ | ✗ | ✗ | ✗ | ✗ | ✗ | ✓ | [4] |
| Rotation | ✓ | ✗ | ✗ | Partial | ✗ | ✗ | ✗ | [5] |
| SSVQ | ✓ | ✗ | ✗ | ✗ | ✓ | ✗ | ✓ | [6] |
| QUIP | ✗ | ✗ | ✓ | Partial | ✗ | ✗ | ✗ | [7] |
| SpinQuant | ✗ | ✓ | ✓ | ✓ | ✗ | ✓ | ✗ | [8] |
| OSTQuant | ✗ | ✓ | ✗ | ✓ | ✗ | ✗ | ✓ | [9] |
| CBQ | ✗ | ✓ | ✓ | ✓ | ✗ | ✗ | ✓ | [10] |
| DuQuant | ✗ | ✓ | ✓ | ✓ | ✗ | ✗ | ✗ | [11] |
| ResQ | ✗ | ✓ | ✓ | ✓ | ✗ | ✗ | ✓ | [12] |
| **Ours** | ✓ | ✓ | ✓ | ✓ | ✓ | ✓ | ✗ | [13] |

[3] Dimension expansion; Hessian-based quantization.

[4] Converts weights to standard Gaussian distribution, similar to QUIP.

[5] Uses rotation and scaling linear transformation to smooth gradient propagation.

[6] Separates sign and absolute value of weights; uses iterative freezing for optimization.

[7] Fixed codebook; exploits sub-Gaussian weight distribution.

[8] Adds Hadamard rotation for low-bit activation and KV cache quantization.

[9] Introduces learnable transformations to enhance quantization space utilization.

[10] Proposes cross-block dependency mechanism to maintain inter-layer relationships.

[11] layer-wise differentiated strategies and novel outlier impact metric

[12] uses rotation and permutation to smooth outlier

[13] combines PCA and rotation to reduce activation outliers

# B ALGORITHM SUPPLEMENT

Our proposed quantization method supports flexible bit-width allocation for each linear layer within a Transformer model. We mention arbitrary bit quantization, more specifically, the combination of bit-widths assigned to different linear layers collectively reflects the overall arbitrary bit quantization effect for the entire model. We allow independent selection of quantization parameters for each layer to realize the desired bit-width configuration. Within the search space formed by these parameter combinations, we seek a suitable solution that balances compression and accuracy. It is important to note that in the current work, we have not optimized the parameter search procedure itself. Instead, we use common grid search techniques to determine the optimal parameter allocation. Our main focus remains on improving the quantization algorithm itself, particularly in handling outliers after quantization.

The quantization workflow proceeds as follows: for each Transformer layer, we sequentially compress its seven linear sublayers. Different quantization parameters can be chosen for each sublayer, resulting in varied compression effects. Within each layer, we select the best-performing configuration for a given compression ratio. For the cross-layer parameter selection, we formulate a grid search problem with the objective of maximizing overall model performance while satisfying a constraint on the total model size. This procedure is summarized in Algorithm 1, which takes as input the original weights, candidate configurations (each with associated compression ratio and performance metric), and a maximum size budget. The algorithm outputs the selected parameters, codebooks, and quantization indices.

---

**Algorithm 1** NuBitQ Compression Parameter Selection for Transformer blocks

---

**Input:** Original weights $W^\star$, candidate configurations $\{(R_{i,s,j}, P_{i,s,j})\}$, size limit SizeMax
**Output:** Selection $x_{i,s,j} \in \{0, 1\}$, codebooks $C_{i,s}$, indices $\mathrm{idx}_{i,s}$

1: Prepare $\{R_{i,s,j}, P_{i,s,j}\}$ for all $i, s, j$.
2: Solve:

$$\max_{x_{i,s,j}} \quad \sum_{i,s,j} P_{i,s,j}\, x_{i,s,j}$$

$$\text{s.t.} \quad \sum_{j} x_{i,s,j} = 1,$$

$$\sum_{i,s,j} R_{i,s,j} \cdot \mathrm{LayerSize}_{i,s} \cdot x_{i,s,j} \le \mathrm{SizeMax}.$$

3: For each $i, s$, find

$$j^\star = \arg\max_{j} x_{i,s,j}.$$

4: Quantize $W_{i,s}^\star$ using NuBitQ with parameters $\theta_{j^\star} = (d, g, r, c, b)$:

Split $W_{i,s}^\star$ into subvectors of dimension $d$,

partition subvectors into $g$ groups; apply scaling,

for each group $k = 1, \ldots, g$ :

build $r$ codebooks $C_1^{(k)}, \ldots, C_r^{(k)}$,

where $C_l^{(k)} \in \mathbb{R}^{c \times d_c}, \quad l = 1, \ldots, r,$

using beam search with width $b$,

assign indices $\mathrm{idx}_{i,s}$ to closest centers in codebooks.

5: Return $\{x_{i,s,j}\}, \{C_{i,s}\}, \{\mathrm{idx}_{i,s}\}$.

---

This algorithm 1 implements the core procedure for selecting quantization parameters across Transformer layers and sublayers. For each linear sublayer, multiple candidate configurations—differing in quantization granularity, codebook size, compression ratio, and other hyperparameters—are prepared. The goal is to select exactly one configuration per sublayer to maximize the overall performance metric (e.g., accuracy) while ensuring the total compressed model size does not exceed a specified budget. The optimization is formulated as a constrained combinatorial selection problem, which we solve via grid search given the manageable parameter space. After selection, the weights are quantized using our NuBitQ method, featuring vector splitting, group-wise scaling, multi-codebook construction via beam search, and index assignment.

To address the accuracy degradation caused by extreme weight values (outliers) that are poorly represented by standard quantization, we introduce an outlier compensation mechanism as a complement to NuBitQ in Algorithm 2 . The procedure begins by quantizing the original weights with NuBitQ to obtain the normal codebook and quantization indices. Then, a sliding window approach scans the weight matrix to identify outliers—elements whose absolute deviation from local statistics (mean and standard deviation) exceeds a predefined threshold. Extracted outliers are clustered separately to form an outlier-specific codebook and corresponding indices. This two-tier approach enables better representation of extreme values without excessively increasing the overall codebook size. Finally, layer-wise compensation is applied based on per-layer outlier statistics. Depending on the outlier ratio of each Transformer block, different compensation strategies are used: from direct mean squared error compensation to parameter tuning via KL-divergence or a global model-based update. The reconstructed weight matrix combines the compensated normal quantized weights with the outlier reconstructions, effectively retaining important information carried by outliers and thus improving the accuracy of the quantized model.

---

**Algorithm 2** NuBitQ-OCP with Outlier Compensation

---

**Input**: Original weights $W^\star$, outlier threshold $\alpha$, sliding window size $w$
**Output**: Normal codebook $Q(E)$, normal indices $Q(I)$, outlier codebook $E_o$, outlier indices $I_o$

 1: Quantize $W^\star$ with NuBitQ to get $Q(E), Q(I)$
 2: Initialize outlier mask $M \leftarrow \mathbf{0}$ with shape of $W^\star$
 3: **for** each sliding window segment $S$ of size $w$ in $W^\star$ **do**
 4:     Compute mean $\mu_S$ and std $\sigma_S$
 5:     **for** each element $w_{ij}$ in $S$ **do**
 6:         **if** $|w_{ij} - \mu_S| > \alpha \cdot \sigma_S$ **then**
 7:             $M_{ij} \leftarrow 1$
 8:         **end if**
 9:     **end for**
10: **end for**
11: Extract outliers $O = \{w_{ij} \mid M_{ij} = 1\}$
12: **if** $|O| > 0$ **then**
13:     Cluster $O$ to obtain outlier codebook $E_o$ and indices $I_o$
14: **else**
15:     $E_o \leftarrow \emptyset, I_o \leftarrow \emptyset$
16: **end if**
17: **for** each Transformer block $i$ **do**
18:     **if** $\beta_i > \theta_2$ **then**
19:         Compute $\Delta W_i^\star$ by Eq. (5)
20:         Update $W_{\mathrm{comp},i} \leftarrow Q(W_i^\star) + \Delta W_i^\star$
21:     **else if** $\theta_1 < \beta_i \leq \theta_2$ **then**
22:         Compute $\theta_i^\star$ by Eq. (6)
23:         Update layer parameters accordingly
24:     **else**
25:         Compute $\theta^\star$ by Eq. (7)
26:         Update global compensation parameters
27:     **end if**
28: **end for**
29: **for** each position $(i, j)$ **do**
30:     **if** $M_{ij} = 1$ **then**
31:         $W_{ij}^{comp} \leftarrow E_o[I_o(i, j)]$
32:     **else**
33:         $W_{ij}^{comp} \leftarrow Q(E)[Q(I)(i, j)]$
34:     **end if**
35: **end for**
36: **return** $Q(E), Q(I), E_o, I_o$

---

## C  ERROR DISTRIBUTION COMPARISON

In this section, we supplement the comparison of error distributions between non-uniform quantization and uniform quantization across various models, as illustrated in Figure 6. This comparison aims to demonstrate that the difference in error distribution due to different quantization methods is not only theoretically supported by previous research (Gong et al., 2024), but also empirically validated through experiments on a diverse set of models.

## D  OUTLIER SCORES ADDENDUM

In this section, we will provide a detailed derivation of our Outlier Scores, compare the computational complexity between the Hessian matrix and our proposed metric, and finally explain the applicability of our metric. It is important to note that our metric is not purely an empirical observation, nor is it merely theoretical speculation. Instead, it integrates both empirical findings and theoretical derivation. Experimental results further demonstrate that our metric aligns well with practical outcomes.

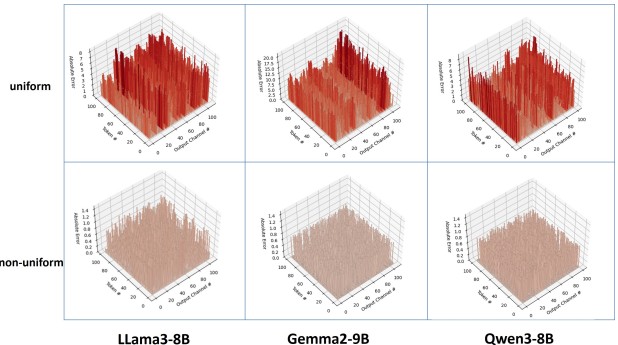

Figure 6: Error distribution between uniform quantization and non-uniform quantization across different models (the analysis is based on the weights of the second layer, quantized using 4-bit precision.).

## D.1 OUTLIER SCORES' THEORETICAL DERIVATION

### D.1.1 NOTATION AND PRELIMINARIES

Consider the $i$-th Transformer layer, which contains $J = 7$ linear sublayers. Denote the original and quantized weight matrices of the $j$-th sublayer as

$$W_{i,j}^{\star} \in \mathbb{R}^{m_j \times n_j}, \quad Q(W_{i,j}) \in \mathbb{R}^{m_j \times n_j},$$

and define the weight perturbation as

$$\Delta W_{i,j} := Q(W_{i,j}) - W_{i,j}^{\star}.$$

Let the final model output be $Y_L$, and the output perturbation caused by quantization be $\Delta Y_L$.

### D.1.2 PERTURBATION PROPAGATION AND JACOBIAN MATRICES

The perturbation $\Delta W_{i,j}$ leads to an output perturbation $\Delta Y_i$ from the $i$-th Transformer layer, which in turn affects the final output $Y_L$. By first-order Taylor expansion, we have the approximation:

$$\Delta Y_L \approx J_{i \to L} \Delta Y_i = J_{i \to L} \sum_{j=1}^{J} J_{i,j}(\Delta W_{i,j}), \tag{8}$$

where:

- $J_{i,j} := \frac{\partial Y_i}{\partial W_{i,j}}$ denotes the local sensitivity (Jacobian matrix) of the $j$-th sublayer's weight perturbation on the layer output perturbation.

- $J_{i \to L} := \frac{\partial Y_L}{\partial Y_i}$ denotes the sensitivity from the $i$-th layer output to the final output.

### D.1.3 EXPECTED SQUARED NORM OF OUTPUT PERTURBATION

We are interested in the average magnitude of output perturbation caused by quantization, measured by the expected squared Frobenius norm:

$$I_i := \mathbb{E}\left[\|\Delta Y_L\|_F^2\right].$$

Substituting Eq. (8) and assuming the perturbations of different sublayers are mutually independent and zero-mean, we obtain

$$I_i = \mathbb{E}\left[\left\|J_{i\to L}\sum_{j=1}^{J}J_{i,j}(\Delta W_{i,j})\right\|_F^2\right] \tag{9}$$

$$= \sum_{j=1}^{J}\mathbb{E}\left[(\Delta W_{i,j})^\top M_{i,j}(\Delta W_{i,j})\right], \tag{10}$$

where the matrix

$$M_{i,j} := \left[J_{i,j}^\top J_{i\to L}^\top J_{i\to L} J_{i,j}\right].$$

Cross terms vanish due to independence.

**On the Assumptions of Independence and Zero Mean** These assumptions are based on the following considerations:

- The 7 sublayers within a Transformer layer are not strictly sequentially connected. Since quantization errors mainly arise independently from each sublayer's weights, it is reasonable to approximate their perturbations as statistically independent.
- The zero-mean assumption corresponds to an unbiased quantization noise model, meaning that on average the quantization does not introduce systematic bias, which simplifies the analysis of second-order effects.

### D.1.4 INTERPRETATION OF $M_{i,j}$

The matrix $M_{i,j}$ jointly captures the sensitivity of sublayer $j$ to weight perturbations and the amplification effect of perturbations propagated through subsequent layers. It thus encodes the complex coupling of local input activations and downstream propagation.

**Decomposition Based on Hessian Trace Technique** The complex second-order matrix $M_{i,j}$ capturing the sensitivity of the loss to weight perturbations in sublayer $(i,j)$ can be approximately decomposed into two interpretable components:

- The statistical properties of the input activations, represented by the covariance matrix

$$C_{i,j} := \mathbb{E}\left[x_{i,j}x_{i,j}^\top\right],$$

where $x_{i,j}$ denotes the input activation to the $j$-th sublayer in layer $i$. This matrix characterizes the dominant directions and energy distribution of the input space relevant to perturbations.

- The amplification effect of subsequent layers, approximated by the product of the squared Frobenius norms of their weights,

$$\prod_{k=i+1}^{L}\|W_k\|_F^2,$$

serving as a coarse estimation of how perturbations propagate and potentially grow or attenuate through the network.

Consequently, according to Hessian trace approximation techniques (Dong et al., 2020), the quadratic form involving weight perturbations $\Delta W_{i,j}$ can be approximated as

$$(\Delta W_{i,j})^\top M_{i,j}\Delta W_{i,j} \approx \|\Delta W_{i,j}\|_F^2 \cdot \mathrm{tr}(C_{i,j}) \cdot \prod_{k=i+1}^{L}\|W_k\|_F^2,$$

where $\|\Delta W_{i,j}\|_F^2$ quantifies the magnitude of the weight perturbation itself, reflecting the intensity of quantization error or deviation in the sublayer.

**Mathematical Approximation of the Decomposition**   By exploiting these independence and low-rank approximations, the quadratic form can be approximated by a scalar product:

$$(\Delta W_{i,j})^\top M_{i,j} \Delta W_{i,j} \approx \|\Delta W_{i,j}\|_F^2 \cdot \mathrm{tr}(C_{i,j}) \cdot \prod_{k=i+1}^{L} \|W_k\|_F^2.$$

Here:

- $\mathrm{tr}(C_{i,j})$ approximates the overall energy or sensitivity of input activations;

- $\prod_{k=i+1}^{L} \|W_k\|_F^2$ approximates the perturbation amplification factor through subsequent layers;

- the original complex high-dimensional matrix $M_{i,j}$ is simplified into a product of two scalar factors.

This decomposition rests on the core idea of Hessian trace techniques: by leveraging the linearity of the trace and expectation, the complicated second-order matrix impact is transformed into a product of input activation statistics and weight amplification effects, enabling computational simplification and clearer physical interpretation.

### D.1.5   DEFINITION OF OUTLIER SCORE

Combining the above, the expected perturbation magnitude of the $i$-th layer can be approximated as

$$I_i \approx \sum_{j=1}^{J} \|\Delta W_{i,j}\|_F^2 \cdot \mathrm{tr}(C_{i,j}) \cdot \prod_{k=i+1}^{L} \|W_k\|_F^2.$$

To avoid numerical instability and convert multiplicative relations into additive ones, we define the outlier score via logarithms:

$$\beta_i := \sum_{j=1}^{J} \left( \log \|\Delta W_{i,j}\|_F^2 + \log \mathrm{tr}(C_{i,j}) + \sum_{k=i+1}^{L} \log \|W_k\|_F^2 \right) \tag{11}$$

This score collectively reflects the perturbation magnitude, input statistical properties, and inter-layer amplification effect, and can be used to identify the modules most vulnerable to outlier perturbations.

### D.2   OUTLIER SCORES' COMPUTATIONAL COMPLEXITY

Consider the weight matrix of a linear sublayer $W \in \mathbb{R}^{m \times n}$, where $m$ is the output dimension and $n$ is the input dimension.

**Hessian Matrix**   The Hessian matrix is the second-order derivative matrix of the loss function with respect to the weights, having size $(mn) \times (mn)$.

- **Storage complexity:** $\mathcal{O}(m^2 n^2)$, which grows quadratically with the number of parameters, resulting in very high memory requirements.
- **Computational complexity:** Direct computation of each second derivative also costs about $\mathcal{O}(m^2 n^2)$. Although Hessian-vector product approximations exist, they still require multiple forward and backward passes and remain computationally expensive.

Therefore, computing and storing the Hessian matrix is typically infeasible for large-scale models.

**Outlier Score**   From the derivation, the key computation in the outlier score involves:

$$(\Delta W)^\top M \Delta W \approx \|\Delta W\|_F^2 \cdot \mathrm{tr}(C) \cdot \prod_{k} \|W_k\|_F^2,$$

where:

- $\|\Delta W\|_F^2 = \sum_{i=1}^m \sum_{j=1}^n (\Delta W_{ij})^2$ can be computed in $\mathcal{O}(mn)$ time.
- The input activation covariance matrix $C = \mathbb{E}[xx^\top] \in \mathbb{R}^{n \times n}$ is estimated by sampling $S$ input activation vectors $\{x^{(1)}, x^{(2)}, \ldots, x^{(S)}\}$, where $S$ denotes the number of sampled activations used to compute the empirical covariance. Computing the full covariance matrix requires $\mathcal{O}(Sn^2)$ operations and $\mathcal{O}(n^2)$ storage.
- However, since only the trace $\mathrm{tr}(C) = \sum_{i=1}^n \mathbb{E}[x_i^2]$ is needed for the outlier score, we only compute and store the diagonal elements (i.e., the average squared activations per input dimension). This reduces the storage complexity to $\mathcal{O}(n)$, and the computation to $\mathcal{O}(Sn)$.
- The product of Frobenius norms of subsequent layers' weights, $\prod_k \|W_k\|_F^2$, involves operations on much smaller matrices and therefore has negligible computational cost compared to Hessian computations.

Hence, the overall computational complexity of the outlier score is approximately

$$\mathcal{O}(mn + Sn + \sum_k m_k n_k),$$

with storage complexity $\mathcal{O}(n)$, which is significantly lower than that of explicitly computing and storing the full Hessian matrix.

**Complexity Comparison** Computing the Hessian matrix is computationally and storage-wise prohibitive in large-scale models due to its quadratic complexity, as summarized in Table 5. In contrast, the outlier score avoids explicit second-order computations by employing approximations, reducing the complexity to simpler norm and covariance calculations. Therefore, the outlier score provides a theoretically meaningful and practically feasible metric for quantifying model sensitivity to perturbations.

|  | Computational Complexity | Storage Complexity |
|---|---|---|
| Hessian Matrix | $\mathcal{O}(m^2 n^2)$ | $\mathcal{O}(m^2 n^2)$ |
| Outlier Score | $\mathcal{O}(mn + Sn + \sum_k m_k n_k)$ | $\mathcal{O}(n)$ |

Table 5: Complexity comparison

### D.3 Outlier Scores' Applicability

Previous studies Lin et al. (2024b) have demonstrated that during model quantization, the importance of the earlier layers is significantly greater than that of the subsequent layers, while the final layer directly impacts the ultimate output performance. As shown in Figure 7, when computing the outlier scores for the Llama3-8B model, our metric shows a clear positive correlation with these findings, assigning higher outlier scores to the early layers and the last layer. This behavior stems from the design of our outlier score: it incorporates not only the quantization errors of the original weights but also the amplification effect of activations on the error. Unlike other methods that focus solely on the influence of activations, our score additionally accounts for the impact of weight perturbations propagating through subsequent layers, which theoretically leads to a stronger bias toward the sensitivity of the earlier layers. In summary, this result reflects a well-founded combination of theoretical derivation and empirical performance of our outlier score, validating its effectiveness and applicability as a metric for estimating layer-wise quantization sensitivity.

## E Experiment Addendum

### E.1 Experimental Setup

**Hardware Environment** All comparative experiments are conducted on a cluster equipped with four NVIDIA A100 GPUs. In the ablation studies, the Ampere architecture corresponds to the A100 GPU, while the Ada Lovelace architecture corresponds to the RTX 4070 GPU.

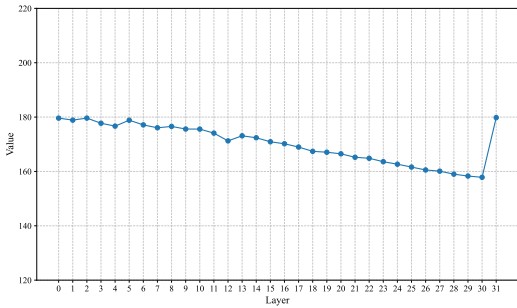

Figure 7: Outlier scores of Llama3-8B with 2-bit quantization across 32 layers.

**Models** The base models include LLaMA3 (Dubey et al., 2024) with 8B and 70B parameters, Qwen3 (Yang et al., 2025) with 8B and 14B parameters, as well as Gemma 2 (Team et al., 2024) with 9B and 27B parameters. These models represent different types and sizes, enabling a comprehensive evaluation of the proposed framework's performance.

**Baselines** We use `FP16` precision as the baseline. For 4-bit quantization, we compare our method with AQLM (Egiazarian et al., 2024), VPTQ (Liu et al., 2024), and GPTVQ (Van Baalen et al., 2024), and conduct similar comparisons for ultra-low-bit quantization. We also evaluate the integration of our OCP into existing methods. The compensation thresholds are set so that 10% of parameters receive linear-level compensation, 10% model-level compensation, and the remaining transformer-level compensation.

**Evaluation Metrics** The evaluation metrics follow LLMCBench (Yang et al., 2024). Perplexity is measured on WikiText2 (Merity et al., 2017) and Penn Treebank(PTB) (Prasad et al., 2014). We comprehensively evaluate three aspects: knowledge ability, reasoning ability, and reliability. Knowledge ability, tested by MMLU (Hendrycks et al., 2021), measures world understanding; reasoning ability, evaluated via QNLI (Rajpurkar et al., 2016) and MNLI (Williams et al., 2018), reflects inference capacity; reliability, assessed with AdvGLUE (Wang et al., 2021) and TruthfulQA, gauges robustness to noise.

### E.2 ADDITIONAL COMPARATIVE EXPERIMENTAL RESULTS

To further verify the effectiveness of our method in ultra-low-bit quantization, we evaluated Qwen3-8B and Gemma2-9B models. We compared different quantization schemes—2-bit and 3-bit—against a 16-bit floating-point baseline. The methods tested include AQLM, VPTQ, GPTVQ, and our approach combined with Optimization Correction Procedure (ours+OCP).

All models were tested on the same datasets covering four knowledge domains: Humanities, STEM, Social Sciences, and Others. We also used downstream tasks (QNLI, MNLI) and trustworthiness metrics (advglu, T.mc1, T.mc2) to assess overall model performance and reliability. Tables 6 and 7 show that 16-bit baselines achieve the best performance. Traditional ultra-low-bit methods like AQLM and VPTQ suffer clear drops in accuracy and trustworthiness, especially below 4 bits.

Our method with OCP consistently outperforms other ultra-low-bit quantization techniques at both 2-bit and 3-bit levels. It achieves accuracy and inference results close to or better than some 4-bit methods, with more stable trustworthiness. For example, at 3-bit quantization, Qwen3-8B reaches 70.26% average knowledge accuracy, higher than others (64.46%-67.60%), and Gemma2-9B hits 71.36%, nearly matching the 16-bit baseline (71.86%). These results confirm our approach effectively compresses large models while maintaining performance, making it suitable for resource-limited deployments.

### E.3 ADDITIONAL ABLATION EXPERIMENTS

### E.3.1 COMPARISON BEFORE AND AFTER USING OCP

This subsection experimentally demonstrates the effectiveness and adaptability of the OCP method.

| Method | #Bits | Knowledge (%) ↑ | | | | | Inference (%) ↑ | | Trustworthiness (%) ↑ | | |
|---|---|---|---|---|---|---|---|---|---|---|---|
| | | Hums. | STEM | Social | Other | Avg. | QNLI | MNLI | advglu | T.mc1 | T.mc2 |
| - | 16 | 63.55 | 72.33 | 83.23 | 76.93 | 72.84 | 89.88 | 71.62 | 58.81 | 35.00 | 55.18 |
| AQLM | 4 | 59.94 | 73.10 | 81.98 | 72.31 | 71.83 | 68.46 | 44.30 | 54.07 | 32.80 | 52.78 |
| VPTQ | 4 | 24.78 | 25.84 | 23.33 | 25.91 | 24.95 | 49.51 | 33.69 | 43.09 | 22.64 | 48.08 |
| GPTVQ | 4 | 61.74 | 72.30 | 82.71 | 75.72 | 71.83 | 89.22 | 68.92 | 52.30 | 34.88 | 54.71 |
| **NuBitQ** | 4 | **62.47** | 73.00 | 82.55 | 77.27 | 72.55 | 87.53 | 69.75 | 55.69 | 35.13 | 55.74 |
| AQLM | 3 | 54.54 | 63.35 | 74.46 | 69.34 | 64.21 | 56.31 | 63.50 | 48.37 | 33.17 | 52.93 |
| AQLM+OCP | 3 | 54.73 | 63.55 | 74.81 | 69.62 | 64.46 | 56.01 | 63.75 | 48.97 | 34.03 | 52.96 |
| VPTQ | 3 | 25.06 | 29.03 | 31.13 | 25.51 | 27.35 | 49.40 | 34.62 | 43.36 | 22.77 | 48.28 |
| VPTQ+OCP | 3 | 32.33 | 37.82 | 37.39 | 32.91 | 35.11 | 50.37 | 41.73 | 46.58 | 31.83 | 48.39 |
| GPTVQ | 3 | 56.85 | 67.00 | 78.91 | 72.67 | 67.52 | 79.25 | 50.71 | 50.68 | 34.15 | 55.44 |
| GPTVQ+OCP | 3 | 57.00 | 67.20 | 79.04 | 72.49 | 67.60 | 75.63 | 52.53 | 51.08 | 34.27 | 55.31 |
| **NuBitQ+OCP** | 3 | **58.17** | **68.59** | **81.02** | **73.26** | **70.26** | **82.49** | **64.43** | **53.17** | **34.55** | **55.75** |
| AQLM | 2 | 53.75 | 59.21 | 73.32 | 66.59 | 62.18 | 79.60 | 52.79 | 52.30 | 30.48 | 50.76 |
| AQLM+OCP | 2 | 52.50 | 58.05 | 72.96 | 66.04 | 61.30 | 79.54 | 56.74 | 51.76 | 30.36 | 50.41 |
| VPTQ | 2 | 26.65 | 25.65 | 24.05 | 24.21 | 25.30 | 50.95 | 32.40 | 45.26 | 22.77 | 49.95 |
| VPTQ+OCP | 2 | 30.37 | 33.23 | 31.78 | 31.09 | 31.62 | 49.23 | 39.01 | 47.82 | 30.03 | 48.26 |
| GPTVQ | 2 | 41.89 | 44.27 | 53.04 | 50.89 | 46.92 | 54.73 | 47.80 | 44.44 | 30.72 | 49.15 |
| GPTVQ+OCP | 2 | 42.64 | 45.00 | 53.30 | 50.89 | 47.39 | 58.20 | 47.27 | 44.17 | 31.21 | 49.51 |
| **NuBitQ+OCP** | 2 | **54.07** | **60.64** | **74.52** | **67.78** | **64.25** | **80.13** | **57.08** | **52.83** | **30.22** | **51.81** |

Table 6: Accuracy of different ultra-low-bit quantization methods on Qwen3-8B across various tasks.

| Method | #Bits | Knowledge (%) ↑ | | | | | Inference (%) ↑ | | Trustworthiness (%) ↑ | | |
|---|---|---|---|---|---|---|---|---|---|---|---|
| | | Hums. | STEM | Social | Other | Avg. | QNLI | MNLI | advglu | T.mc1 | T.mc2 |
| - | 16 | 64.99 | 65.44 | 83.49 | 76.74 | 71.86 | 87.36 | 68.61 | 60.16 | 43.33 | 60.69 |
| AQLM | 4 | 65.14 | 64.55 | 82.71 | 75.72 | 71.31 | 85.12 | 69.56 | 59.21 | 42.72 | 60.84 |
| VPTQ | 4 | 24.14 | 28.83 | 31.13 | 24.98 | 26.88 | 50.60 | 31.29 | 43.90 | 23.26 | 49.13 |
| GPTVQ | 4 | 65.19 | 64.21 | 83.33 | 76.43 | 71.55 | 87.07 | 69.90 | 58.67 | 43.05 | 60.42 |
| **NuBitQ** | 4 | 64.74 | 65.34 | 83.49 | 76.74 | 71.75 | 87.38 | 67.93 | 60.57 | 43.21 | 60.61 |
| AQLM | 3 | 62.42 | 62.69 | 80.40 | 74.06 | 69.14 | 87.18 | 70.27 | 56.78 | 40.64 | 58.55 |
| AQLM+OCP | 3 | 62.47 | 62.72 | 80.47 | 74.09 | 69.17 | 87.33 | 70.29 | 57.05 | 41.13 | 58.67 |
| VPTQ | 3 | 24.17 | 28.86 | 31.07 | 25.05 | 26.89 | 50.60 | 31.29 | 43.09 | 23.01 | 48.66 |
| VPTQ+OCP | 3 | 30.99 | 31.49 | 37.68 | 33.04 | 33.30 | 48.16 | 38.50 | 47.81 | 39.94 | 53.90 |
| GPTVQ | 3 | 61.91 | 61.99 | 80.89 | 75.29 | 69.18 | 88.03 | 61.04 | 59.49 | 42.11 | 59.83 |
| GPTVQ+OCP | 3 | 61.90 | 62.15 | 81.05 | 75.48 | 69.31 | 88.01 | 61.00 | 59.08 | 41.99 | 59.92 |
| **NuBitQ+OCP** | 3 | **64.19** | **63.29** | **82.77** | **75.17** | **71.36** | **87.34** | **71.98** | **59.73** | **42.38** | **59.91** |
| AQLM | 2 | 57.96 | 55.24 | 74.13 | 69.12 | 63.50 | 86.75 | 69.72 | 48.51 | 41.49 | 58.03 |
| AQLM+OCP | 2 | 58.04 | 55.47 | 74.33 | 69.15 | 63.62 | 86.81 | 70.28 | 49.05 | 41.98 | 58.30 |
| VPTQ | 2 | 24.17 | 28.86 | 31.07 | 25.05 | 26.89 | 50.60 | 31.29 | 43.09 | 23.01 | 48.66 |
| VPTQ+OCP | 2 | 29.76 | 27.09 | 31.81 | 31.75 | 30.10 | 46.62 | 35.99 | 47.28 | 38.23 | 54.37 |
| GPTVQ | 2 | 51.92 | 50.60 | 68.90 | 63.88 | 58.12 | 56.27 | 45.57 | 44.99 | 38.56 | 55.97 |
| GPTVQ+OCP | 2 | 51.05 | 50.20 | 68.61 | 63.94 | 57.69 | 53.72 | 44.22 | 45.53 | 39.17 | 55.86 |
| **NuBitQ+OCP** | 2 | **59.35** | **56.94** | **74.63** | **70.20** | **65.28** | **85.23** | **70.49** | **53.22** | **41.25** | **58.95** |

Table 7: Accuracy of different ultra-low-bit quantization methods on Gemma2-9B across various tasks.

In the experiment, we apply 2-bit quantization only to the first layer's Linear sublayer weights of the Transformer model, keeping other layers at 16-bit precision. After quantization, outlier scores for weights and activations are computed, and OCP compensation is applied based on the outlier codebook pool. Evaluation on the validation set uses perplexity as the metric, analyzing changes before and after quantization at both sublayer and sample levels.

Figures 8a and 8b compare sublayer perplexity before and after OCP, showing that OCP stabilizes perplexity and prevents abnormal spikes caused by quantization errors. Figures 8c and 8d show that OCP reduces perplexity variance across individual samples, further confirming its effectiveness in improving model stability and performance.

Our method identifies outliers via outlier scores, which are key sources of quantization error. We build an outlier codebook pool with multiple compensation codebooks learned from calibration data

to correct various outlier types. The compensation uses a sliding greedy selection over parameter or activation tensors, selecting compensation vectors that minimize local quantization error iteratively, achieving fine-grained correction while balancing accuracy and computation.

The compensation ratio at different granularities can be flexibly adjusted by thresholds. By default, we use 10% fine-grained and 10% coarse-grained compensation, which works well in practice. If needed, one can start with full fine-grained compensation and relax thresholds to find better settings, with minimal extra computational cost.

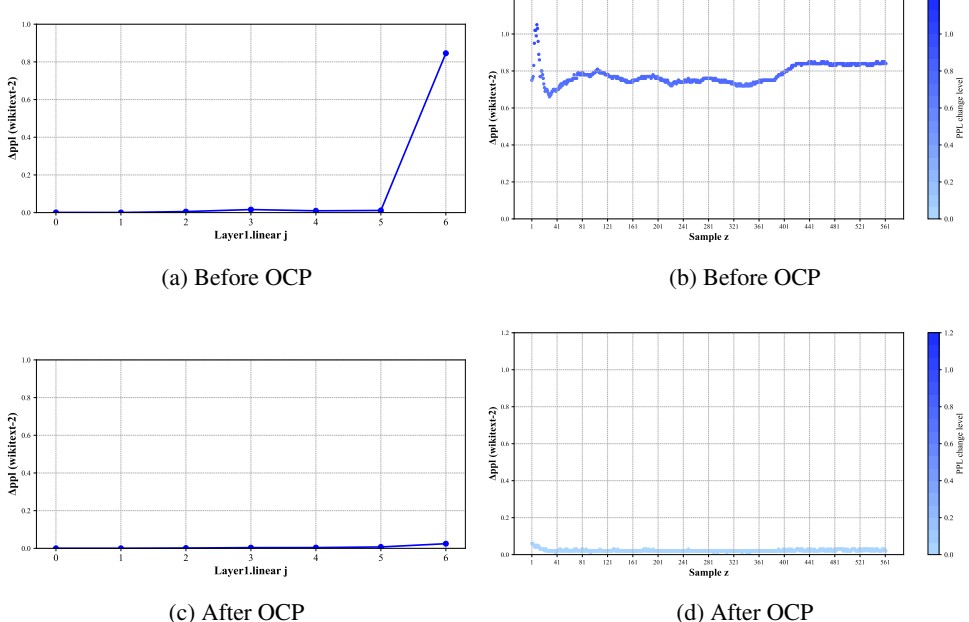

Figure 8: Perplexity comparison at different stages: before/after OCP compensation.

### E.3.2 COMPARISON BETWEEN FINE-TUNING AND OCP

After applying 3-bit quantization to the Transformer layer 0 of the Llama3-8B model, we evaluated two categories of recovery methods: fine-tuning and compensation based on OCP. The fine-tuning approaches included layer-wise linear fine-tuning and layer-wise Transformer fine-tuning, while the OCP methods involved compensation at the linear layer level, Transformer layer level, and the entire model level. It is important to note that full model fine-tuning was not considered in this experiment, as it falls under quantization-aware training, whereas our focus here is on lightweight strategies to recover accuracy post-quantization.

Regarding the experimental setup, the OCP compensation utilized activation data from 50 WikiText-2 samples as reference, following the methodology described in the main text. For fine-tuning, the initial learning rate was set to $10^{-5}$, and 500 WikiText-2 samples were used to ensure adequate yet efficient fine-tuning. The final evaluation was conducted on the WikiText-2 validation set using perplexity as the performance metric to assess the impact of quantization and the effectiveness of recovery methods.

Table 8 presents the perplexity results for the different recovery strategies applied to the quantized Transformer layer 0. It is evident from the results that OCP compensation consistently achieves lower perplexity than fine-tuning, whether applied at the linear layer level or the Transformer layer level, indicating a more effective and efficient recovery of quantization-induced accuracy degradation. Notably, the OCP compensation applied at the whole-model level also outperforms fine-tuning, demonstrating strong global adjustment capabilities. Furthermore, fine-tuning requires significantly more training data and computational resources compared to OCP compensation, which only needs a small set of activation samples, highlighting its superior cost-efficiency.

|  | before | linear | transformer | model |
|---|---|---|---|---|
| **Fine-tuning** | 5.6269 | 5.6176 | 5.6217 | - |
| **OCP** | | 5.5918 | 5.6014 | 5.61371 |

Table 8: Perplexity comparison of fine-tuning and OCP compensation after 3-bit quantization (lower is better).

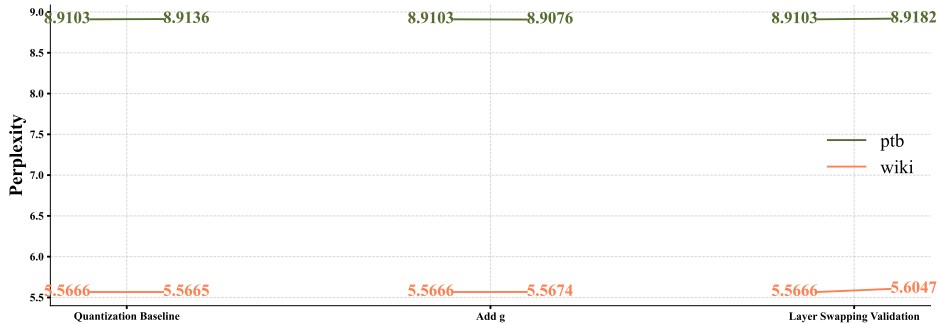

Figure 9: Analysis of unexpected results.

### E.3.3 RANDOMNESS ASSESSMENT OF PENTAGRAM ANOMALIES

To verify the "unexpectedly good" result observed in the main text's hyperparameter abla-tion—where the perplexity after quantization was surprisingly lower than before quantization—we conducted a thorough investigation from three perspectives:

First, we examined whether this phenomenon occurs consistently across different datasets under the same quantization setting. Second, we increased the number of groups $g$ (both theoretically and empirically, increasing $g$ should lead to an increase in perplexity after quantization). If perplexity continues to decrease as $g$ increases, it would suggest that the phenomenon is not accidental. Fi-nally, we applied the same hyperparameter configuration to other layers with the same shape. If the "unexpectedly good" result also appears in other layers, it could further rule out randomness.

The experimental results are shown in Figure 9. Here, "Quantization Baseline" represents the per-plexity results of the original hyperparameter setting that produced the anomaly; "Add $g$" corre-sponds to the results after increasing the number of groups $g$ only; "Layer Swapping Validation" refers to applying the hyperparameters to different layers. Each of the six line segments has its left endpoint representing perplexity before quantization and the right endpoint representing perplexity after quantization.

We observe that the "unexpectedly good" result does not reproduce across different datasets, with increased $g$, or in different layers. In all other cases, perplexity increases after quantization as expected. This indicates that the anomalous improvement is due to a specific combination of dataset, layer, and quantization parameters, and is thus a coincidental outcome rather than a generalizable effect.

## F REPRODUCIBILITY STATEMENT

We have made every effort to ensure the reproducibility of our results. All datasets used in our exper-iments are publicly available. The source code to reproduce our experiments will be released anony-mously as supplementary material, including detailed instructions for data preprocessing, model training, and evaluation. Hyperparameters, model architectures, and training settings are provided in the Appendix and the readme.txt file. Additional implementation details are also described in the supplementary materials.

## G   LLM Usage Statement

We used large language models (LLMs) to aid and polish the writing of this paper. All substantive scientific contributions and ideas are our own.

