# OpenReview forum: "No outlier channels but with outlier blocks"
_ICLR.cc/2026/Conference — ICLR 2026 Poster_

### Official Review · Reviewer_jNs7 · 2025-10-28

**Soundness:** 2
**Presentation:** 1
**Contribution:** 3
**Rating:** 4
**Confidence:** 4

**Summary:**

This work studies the (1) Non-uniform quantization, (2) outlier impact, and (3) outlier compensation by minimization. Overall, the ultimate goal of this work is to push the performance boundary in non-uniform quantization. However, I found the organization of this work to be unclear and missing.

**Strengths:**

+ Thorough evaluation of non-uniform quantization across various models and benchmarks.

**Weaknesses:**

- The figures in this paper are not informative. I cannot understand them if not reading the papers.

- I have several concerns regarding OCP. First, the final OCP methodology seems like the standard calibration with different levels of granularity. This resembles the AQLM's multi-phase optimization that first tackles the layer-wise quantization and then tackles the inter-layer error. Conceptually, there is nothing new about OCP, given that it has been extensively applied in dozens of PTQ works.

- Moreover, the OCP method seems to be compatible with any quantization algorithms, not just non-uniform quantization. If this is the case, then the authors need to compare a lot of uniform baselines.

- Third, I did not understand the necessity of Section 3.2.2. I have checked Appendix D for the detailed derivation of the outlier score. However, I should emphasize that the derivation is definitely not rigorous. The logarithm makes no sense to add. To me, this is just to prevent the exploding effect of the subsequent layers' weight norm. The Jacobian should contain the gradients of na on-linear function, while this work directly ignores them. Furthermore, this outlier score has nothing to do with outliers; it is just a metric to analyze importance. Again, this score seems to be applicable to all kinds of parameters and activations, not just outliers. I could not see how this is *outlier-centric*.

**Questions:**

I am curious to see if this non-uniform weight-only quantization can be accelerated on the modern LLM inference engine such as vLLM and SGLang.

---

> ### Author Response · Authors · 2025-11-24
>
> Thank you for your detailed and candid evaluation, as well as for acknowledging our experimental setup. Your valuable concerns regarding the novelty and theoretical derivation of the OCP method are greatly appreciated. We will provide clearer explanations and additional elaborations on these aspects in the revised manuscript to enhance the logical rigor and clarity of our paper. Below are our responses and the improvements made:
>
>
>
> ## **W1: Data Presentation**
> In Figure 3, we reveal the existence of outliers, but it seems the reviewers would like us to provide a deeper analysis: specifically, to explain how the errors develop into outliers rather than just presenting the results. If a more detailed analysis can make the paper’s logic clearer, we are more than willing to add this analysis process.
>
> Regarding our theory that "non-uniform quantization contains outliers that differ from those in uniform quantization," we refer to the uniform quantization error definition in [1]. Large language models (LLMs) contain so-called "outlier channels," meaning some feature dimensions have values much larger than others. In our paper, Figures 1 and 5 demonstrate that the quantization errors from non-uniform quantization do not present in a channel-wise manner.
>
> To explain how the non-uniform quantization error develops into outliers, we define "outliers" as errors that strongly impact accuracy. We illustrate this with an example showing how the error evolves into an outlier: Although the overall error within a block is small, significant errors can still appear inside the block. Suppose the dimension \(d=4\), and after replacing four vectors with codebook vectors, the quantization error matrix is:
>
> $$
> \begin{bmatrix}
> 0.3118336 & 0.0368802 & 0.02044559 & 0.09176590
> \end{bmatrix}
> $$
> $$
> \begin{bmatrix}
> 0.5819435 & 1.1185487 & 0.07737651 & 0.06973236
> \end{bmatrix}
> $$
> $$
> \begin{bmatrix}
> 0.0118336 & 0.0300486 & 0.01504153 & 0.27175903
> \end{bmatrix}
> $$
> $$
> \begin{bmatrix}
> 0.3368979 & 0.0118548 & 1.10819237 & 0.00512973
> \end{bmatrix}
> $$
>
>
>
>
> The activation vectors are:
>
> $$
> \begin{bmatrix}
> 0.4675 \\
> 0.8790 \\
> 0.6623 \\
> 0.8743
> \end{bmatrix}
> $$
>
> Resulting in output errors of:
>
>
> \begin{bmatrix}
> 0.2719\\
> 1.3674 \\
> 0.2794 \\
> 0.9063
> \end{bmatrix}
>
> Although the overall non-uniform quantization error seems low due to clustering, some vectors exhibit large deviations to fit certain codebook vectors. This causes the output error to grow significantly. While the average error per dimension appears similar, the actual output errors vary considerably.
>
> Moreover, these sporadic large errors are not only related to the clustering quality within the current block but are also amplified by activations and propagation across layers, which can cause the final error to be hundreds of times larger than the original small error. This amplification effect is exactly why we designed the outlier scoring mechanism.
>
>
>
>
> ## **W2: Novelty of OCP**
> Our initial idea was to follow a simple and naive approach, similar to AWQ (currently one of the most popular quantization methods), aiming to solve the problem with minimal cost. Therefore, when we discovered the characteristics of outliers caused by non-uniform quantization, we began to analyze and address them.
>
> The OCP method indeed operates at different granularities, but it fundamentally differs from AQLM’s standard calibration approach: AQLM does not study outliers explicitly; instead, it performs layer-wise fine-tuning by collecting a calibration dataset to obtain an optimized result. As we demonstrate experimentally in the appendix, OCP provides a more fine-grained and targeted compensation for outliers, whereas AQLM adjusts the entire transformer block as a whole. Moreover, OCP does not involve backpropagation, which means its quantization overhead is significantly lower than that of AQLM.
>
> Additionally, in Appendix E.3.2, we show that the accuracy improvement from layer-wise fine-tuning is inferior to that achieved by OCP.
>
> |                   | before  | linear  | transformer | model   |
> |-------------------|---------|---------|-------------|---------|
> | **Fine-tuning**   | 5.6269  | 5.6176  | 5.6217      |   -     |
> | **OCP**           |     -    | 5.5918  | 5.6014      | 5.61371 |
>
> *Table: Perplexity comparison of fine-tuning and OCP compensation after 3-bit quantization (lower is better).*

---

> ### Author Response · Authors · 2025-11-24
> **for W3 and W4**
>
> ## **W3: Compatibility of OCP with Uniform Quantization**
>
> OCP can be adapted to any transformer model. Regarding the compatibility of OCP with uniform quantization, we believe it is feasible but not strongly necessary.
>
> This is because OCP is specifically designed to optimize outliers caused by non-uniform quantization, and the outlier characteristics differ between uniform and non-uniform quantization due to their distinct quantization methods. Outliers in uniform quantization, often caused by clipping, can usually be mitigated by channel-wise scaling or other techniques. In contrast, outliers from non-uniform quantization tend to be more dispersed, so OCP’s fine-grained approach is better suited to alleviate these. For uniform quantization, OCP may only provide a compensatory effect without targeted optimization.
>
> Additionally, Reviewer 7kTW suggested that we compare OCP with other optimization components, such as rotation-based methods, which are more suitable for uniform quantization. We have conducted experiments accordingly and will present the results to further support our viewpoint.
>
> | **Method**              | **Smooth** | **Fine-tune** | **Rotation** | **OCP** | **Time Cost(s)** | **Memory Cost(kb)** | **PPL**  |
> |-------------------------|------------|---------------|--------------|---------|---------------|-----------------|----------|
> | **before optimization**              |            |               |              |         | 0             | 0               | 5.6269   |
> |  | ✓          |               |              |         | 2.75          | 1.188           | 5.6244   |
> |                         |            | ✓             |              |         | 1622.6        | 0               | 5.6217   |
> |                         |            |               | ✓            |         | 84.58         | 448.8           | 5.6201   |
> |                         |            |               |              | ✓       | 45.17         | 114.7           | 5.6014   |
> |                         | ✓          |               |              | ✓       | 46.33         | 115.9           | 5.6002   |
> |                         |       | ✓             |              | ✓       | 1570.1        | 114.7           | 5.6101   |
> |                         |       |          | ✓            | ✓       | 119.05        | 573.4           | 5.5997   |
>
> *Table: Ablation Study of Various Anomaly Optimization Methods in Non-Uniform Quantization for Llama3-8B*
>
> ---
>
> ## **W4: Necessity of Section 3.2.2**
>
> Regarding the comment that "logarithmic operations are meaningless": the use of logarithms is based on practical experience to ensure usability in real-world applications. Because the parameters involved in the metric calculation are extremely large, weight norm explosions can occur, which renders direct computation meaningless without logarithmic scaling.
>
> Regarding the point that "the Jacobian matrix should include gradients of nonlinear functions": we simplify the derivation to avoid computational complexity and instability. Strictly including nonlinearities would make the Jacobian extremely complex — not only would its dimension be enormous, but it would also depend on specific inputs, making the subsequent expectation and covariance estimations very difficult, especially for large-scale models.
>
> We make an empirical assumption that the linear parts dominate: in deep networks like Transformers, it is observed that perturbations in linear weights tend to have a greater impact on the output. Many nonlinear activations (such as ReLU or GELU) can be locally approximated as linear or have relatively simple gradient distributions, so neglecting nonlinear terms in a first-order approximation does not cause significant error.
>
> Moreover, when computing layer-by-layer, the output of the previous layer after activation becomes the input to the next layer, implicitly incorporating nonlinear transformations.
>
> Our analysis goal differs: we aim to quantify an outlier metric rather than derive an extremely precise perturbation propagation formula. This simplification greatly improves the theoretical practicality and computability. Notably, prior work on linear quantization outliers also ignores activations in their analysis.
>
> Regarding the point that "the outlier score is unrelated to the outlier itself": we have discussed in W1 why errors develop into outliers. About the comment that "the score seems to apply to all types of parameters and activations," there exist previous studies on parameter sensitivity [1][2]. For example, [1] uses the Hessian matrix to evaluate parameter sensitivity. The Hessian is also widely used for mitigating quantization outliers because outliers correspond to highly sensitive parameter errors, not just the parameters themselves. Our evaluation and calculation focus on errors caused by non-uniform quantization, which is essentially different from pure parameter sensitivity analysis.

---

> > ### Author Response · Authors · 2025-11-24
> > **for Q1**
> >
> > # Q1: Acceleration in LLM Inference Engines
> >
> > vLLM does not directly integrate or specifically support lookup-table-based non-uniform quantization kernels. However, other inference frameworks do support similar non-uniform quantization acceleration and have demonstrated notable performance improvements. Large language model inference is typically memory-bound, and non-uniform quantization reduces computational workload through lookup tables, thereby lowering computational pressure and improving inference efficiency. Therefore, even without relying on vLLM, our method can still achieve acceleration in environments that support non-uniform quantization.
> >
> > Currently, many low-level optimizations for quantization have emerged, such as ExLlamaV2 and Marlin. In particular, Marlin combines lookup table methods with GEMM operations. For example, [3] addresses memory bandwidth limitations and challenges posed by low-bit quantization by offline reconstructing quantized weight matrices, vectorizing lookup tables, and employing streaming K-work decomposition, thereby accelerating the inference process.
> >
> > Thank you very much for your attention to our work and your valuable suggestions. Your feedback plays an important role in helping us improve our research. We look forward to your continued guidance and constructive comments, which will greatly contribute to the advancement of this field. We sincerely hope that our revisions meet your expectations and positively reflect in your evaluation.
> >
> > ---
> >
> > ## References
> >
> > [1] S. Kim, C. R. C. Hooper, A. Gholami, Z. Dong, X. Li, S. Shen, M. W. Mahoney, and K. Keutzer, "SqueezeLLM: Dense-and-Sparse Quantization," in *Forty-first International Conference on Machine Learning*, 2024. [Online]. Available: https://openreview.net/forum?id=0jpbpFia8m
> >
> > [2] G. Lee, J. Lee, S. Hong, M. Kim, E. Ahn, D.-S. Chang, and J. Choi, "RILQ: Rank-Insensitive LoRA-Based Quantization Error Compensation for Boosting 2-Bit Large Language Model Accuracy," in *AAAI*, 2025, pp. 18091–18100. [Online]. Available: https://doi.org/10.1609/aaai.v39i17.33990
> >
> > [3] Guo, H., Brandon, W., Cholakov, R., Ragan-Kelley, J., Xing, E. P., & Kim, Y. (2024). Fast matrix multiplications for lookup table-quantized LLMs. *Findings of the Association for Computational Linguistics: EMNLP 2024*, 12419–12433. https://doi.org/10.18653/v1/2024.findings-emnlp.724

---

> ### Author Response · Authors · 2025-11-27
>
> Dear Reviewer jNs7,
>
> I hope this message finds you well! With just under three days remaining in the discussion period, we wanted to kindly check if we have addressed all of your valuable comments and suggestions. If you have any further thoughts or additions, please feel free to share them with us. Your feedback is truly important and greatly helps us improve our work.
>
> Thank you sincerely for your careful review and insightful guidance. We look forward to hearing any further recommendations you might have!

---

> > ### Comment · Reviewer_jNs7 · 2025-11-27
> > **Reply to Rebuttal**
> >
> > I'd like to thank the authors for their detailed response. Regarding comparison to uniform quantization, I agree that OCP has its own target scenario, and comparison of uniform quantization is not needed at this time.
> >
> > Regarding Section 3.2.2, I maintain my opinion that the score metric is too primitive, not rigorous. The author's logic of the score is: Output perturbation -> Expand with Jacobian -> Approximate Jacobian with multiples of weight norm -> bound it with logarithm since it is too large.
> >
> > To me, this logic is not mathematically rigorous. A simple explanation is that if you need to bound your estimation with a logarithm, then it is simply not a valid estimation! Here's why:
> >
> > 1. The authors use multiples of the weight norm and ignore the non-linearity. The authors explained that *"previous layer after activation becomes the input to the next layer, implicitly incorporating nonlinear transformations"*.  However, it only propagates one layer of non-linearity. You still view the current layer to the last layer as a linear model.
> >
> > 2. The authors state that "Many nonlinear activations (such as ReLU or GELU) can be locally approximated as linear or have relatively simple gradient distributions, so neglecting nonlinear terms in a first-order approximation does not cause significant error." This is so wrong. The non-linear activations can never be approximated in linear because of the negative range where the derivative is 0. Besides, when you talk about Transformer, you have to consider Attention and Gate functions; none of those produce linear gradients. The authors essentially approximate transformers with linear networks.
> >
> > 3. "we aim to quantify an outlier metric rather than derive an extremely precise perturbation propagation formula. This simplification greatly improves the theoretical practicality and computability." There are a lot of ways to precisely get the desired metric. For example, if you think the net output-> current layer Jacobian is too large, you can expand the loss perturbation with Fisher, so that you can directly utilize the auto-grad to calculate the metric.
> >
> > Finally, when I was checking the paper, the authors states that "Following Hessian trace approximation techniques [HAWQ-V2], we decompose Eq. (3) into three interpretable components". But I could not find this approximation in the cited paper.

---

> ### Author Response · Authors · 2025-11-28
>
> Dear Reviewer,
>
> Thank you for your recognition of our point that "OCP has its own target scenario." We would like to address each of your concerns in detail:
>
> 1. Regarding the comment that **"You still view the current layer to the last layer as a linear model."** this is not the case. The input to the nth layer is the output of the previous n-1 layers, which means the nonlinearities of all these preceding layers are indeed propagated. We have conducted exploratory experiments comparing the effect of propagating only one layer's nonlinear activation versus propagating all prior nonlinear activations.
> For example, when compensating the 32nd layer of the LLaMA3-8B model after 2bit-quantization , we compared the effects of using only the original activation from layer 31 (one layer of nonlinearity) versus using the quantized activations from all preceding 31 layers (all nonlinearities) on the Wikitext-2 dataset:
> | Quantization Method | Metric Value  |
> |---------------------|--------------|
> | One-layer nonlinearity | 6.432490777 |
> | All nonlinearities     | 6.420929737 |
>
> The results show that considering all preceding nonlinear activations slightly outperforms using only one layer, indicating that propagating multiple nonlinearities helps improve compensation accuracy and that our method partially accounts for multi-layer nonlinear effects.
>
> 2. Regarding the concern that **"the authors essentially approximate transformers with linear networks."** we clarify that we do not treat the Transformer as simply linear. While each block’s overall quantization error magnitude is similar, the impact of the error is uneven across blocks. We assign different anomaly scores to each block and apply block-wise compensation accordingly to reduce the overall error impact. Hence, the linearity observed is only apparent. Moreover, study [1] establishes a linear theorem relating per-layer errors to final perplexity, further supporting our approach of connecting local and global errors.
>
> 3. Regarding the suggestion of **“you can expand the loss perturbation with Fisher”**, we surveyed major Hessian simplification methods, including Hessian trace approximations [2][3][4] and Fisher information-based methods [5][6][7]. We systematically compared these approaches in terms of computational cost and approximation accuracy, and combined with insights from [8], found that diagonal approximations often achieve satisfactory practical results. Therefore, we ultimately chose the trace-based approximation by using the trace of the input activation covariance matrix to measure overall scale. In practical quantization, the Hessian can often be approximated by the square of the activation matrix, making this approach a concise and efficient way to estimate weight perturbation sensitivity on the output.
>
> Finally, regarding the **“could not find this approximation in the cited paper.”**, we followed the methodology of the HAWQ series: HAWQ-V1 uses the maximum Hessian eigenvalue as a sensitivity metric; HAWQ-V2 approximates the full eigenvalue spectrum to replace the entire matrix, offering more comprehensive but computationally expensive sensitivity information. However, the Hessian diagonal elements typically well reflect key characteristics, as verified by our prior research. Based on this, we selected the trace-based approximation using the trace of the input activation covariance matrix. This method efficiently estimates sensitivity with low computational cost, since the Hessian matrix can usually be approximated by the squared activation matrix in quantization tasks.
>
> Based on the above clarifications and additions, we hope the issues have been resolved. If you have any questions or would like to discuss further, we are more than happy to engage. We sincerely hope you acknowledge our improvements and provide a positive review of our manuscript. Your support will greatly encourage us to continue advancing this research.
>
> Best regards,
> The Authors

---

> ### Author Response · Authors · 2025-11-28
>
> ### Reference
>
> [1]Malinovskii, V., Panferov, A., Ilin, I., Guo, H., Richtárik, P., & Alistarh, D. (2025). **HIGGS: Pushing the Limits of Large Language Model Quantization via the Linearity Theorem**. In L. Chiruzzo, A. Ritter, & L. Wang (Eds.), *Proceedings of the 2025 Conference of the Nations of the Americas Chapter of the Association for Computational Linguistics: Human Language Technologies (Volume 1: Long Papers)* (pp. 10857–10886). Association for Computational Linguistics.
> [https://doi.org/10.18653/v1/2025.naacl-long.543](https://doi.org/10.18653/v1/2025.naacl-long.543)
> [https://aclanthology.org/2025.naacl-long.543/](https://aclanthology.org/2025.naacl-long.543/)
>
> [2]Lui, H. W., & Neftci, E. (2021). *Hessian Aware Quantization of Spiking Neural Networks*. In *International Conference on Neuromorphic Systems 2021* (Article 12, pp. 1–5). Association for Computing Machinery. [https://doi.org/10.1145/3477145.3477158](https://doi.org/10.1145/3477145.3477158)
>
> [3]Dong, Z., Yao, Z., Cai, Y., Arfeen, D., Gholami, A., Mahoney, M. W., & Keutzer, K. (2019). *HAWQ-V2: Hessian Aware trace-Weighted Quantization of Neural Networks*. arXiv preprint arXiv:1911.03852. [https://arxiv.org/abs/1911.03852](https://arxiv.org/abs/1911.03852)
>
> [4]Huang, Z., Han, X., Yu, Z., Zhao, Y., Hou, M., & Hu, S. (2025). Hessian-based mixed-precision quantization with transition aware training for neural networks. *Neural Networks*, 182, 106910. https://doi.org/10.1016/j.neunet.2024.106910
> [https://www.sciencedirect.com/science/article/pii/S0893608024008396](https://www.sciencedirect.com/science/article/pii/S0893608024008396)
>
> [5]Wang, F., Yin, H., Zhuang, S., Zhu, H., Li, Y., Qian, L., Zhang, C., Zhao, H., Qian, H., & Li, C. (2025). Efficiently Access Diffusion Fisher: Within the Outer Product Span Space. In *Proceedings of the Forty-second International Conference on Machine Learning*.
> Retrieved from [https://openreview.net/forum?id=b3xzkfd0G1](https://openreview.net/forum?id=b3xzkfd0G1)
>
> [6]Wu, Z., Wang, S., Zhang, J., Chen, J., & Wang, Y. (2025). FIMA-Q: Post-Training Quantization for Vision Transformers by Fisher Information Matrix Approximation. arXiv preprint arXiv:2506.11543. https://doi.org/10.48550/arXiv.2506.11543
> [https://arxiv.org/abs/2506.11543](https://arxiv.org/abs/2506.11543)
>
> [7] Xie, Z., Ma, Y., Zheng, X., Chao, F., Sui, W., Li, Y., Li, S., & Ji, R. (2025). Automated Fine-Grained Mixture-of-Experts Quantization. In *Findings of the Association for Computational Linguistics: ACL 2025* (pp. 27024–27037). Vienna, Austria: Association for Computational Linguistics. https://doi.org/10.18653/v1/2025.findings-acl.1386
> [https://aclanthology.org/2025.findings-acl.1386/](https://aclanthology.org/2025.findings-acl.1386/)
>
> [8]Li, Y. X., Dangel, F., Tam, D., & Raffel, C. (2025). Fishers for Free? Approximating the Fisher Information Matrix by Recycling the Squared Gradient Accumulator. In *Proceedings of the Forty-second International Conference on Machine Learning*.
> Retrieved from [https://openreview.net/forum?id=m3zrHhiCCj](https://openreview.net/forum?id=m3zrHhiCCj)

---

### Official Review · Reviewer_R323 · 2025-10-31

**Soundness:** 3
**Presentation:** 3
**Contribution:** 3
**Rating:** 6
**Confidence:** 4

**Summary:**

This paper addresses the challenges of quantizing LLMs, particularly focusing on the limitations of non-uniform quantization methods. The authors argue that while non-uniform quantization reduces overall error compared to uniform methods, it creates novel and less-understood outlier patterns that are not concentrated in specific channels but rather distributed across what they term "outlier blocks." To tackle this, they propose a two-part framework. The first part is NuBitQ, a flexible non-uniform quantization scheme supporting arbitrary bit-widths and layer-specific configurations without relying on costly Hessian computations. The second contribution is the OCP, a module designed to mitigate performance degradation from these outlier blocks. The OCP is guided by a novel, efficiently computed outlier score that approximates the impact of quantization error by considering weight perturbation, activation statistics, and error propagation through subsequent layers. Based on this score, OCP applies a hierarchy of compensation strategies at the sub-layer, block, and model levels. The authors conduct extensive experiments on LLaMA3, Qwen, and Gemma models, demonstrating that their framework achieves state-of-the-art performance, especially in ultra-low-bit settings.

**Strengths:**

1.The paper provides a valuable analysis of how outlier patterns differ between uniform and non-uniform quantization. The identification of "outlier blocks" rather than just "outlier channels" as the primary challenge in non-uniform quantization is a key insight that effectively motivates the need for new compensation strategies.

2.The Outlier Compensation Plugin is designed as a modular, plug-and-play component. This highlights its general utility.

3.The paper presents comprehensive experimental results across multiple model families and sizes. The framework demonstrates near-lossless performance at 4-bit and shows particularly strong gains in the challenging ultra-low-bit regimes, outperforming state-of-the-art baselines in both perplexity and downstream task accuracy.

**Weaknesses:**

1.The paper claims that OCP is a generalizable, "plug-and-play" module. While perplexity results in Table 1 support this, the downstream task accuracy results in Table 2 present a more mixed picture. When OCP is added to AQLM and GPTVQ, the average MMLU score shows negligible improvement or even a slight decrease. This suggests that the OCP's effectiveness may be dependent on the underlying quantization scheme, potentially conflicting with the claim of broad applicability.

2.The complete NuBitQ-OCP framework involves a considerable number of hyperparameters, including the quantization parameters, the number of compensation levels, and the thresholds that govern the selection of compensation strategies. Although the paper includes some ablation studies, it lacks a broader discussion on the sensitivity and practical guidance for setting these parameters, especially the compensation thresholds, which appear crucial for balancing performance and overhead. This complexity may hinder the method's ease of adoption.

**Questions:**

1.The results in Table 2 indicate that applying OCP to AQLM and GPTVQ does not consistently improve, and sometimes slightly degrades, performance on downstream tasks like MMLU. Why does OCP fail to provide a clear benefit in these cases, unlike the significant perplexity improvements shown in Table 1? Does this imply that OCP is specifically tuned for the error patterns produced by NuBitQ or that it might conflict with the optimization strategies already present in methods like AQLM and GPTVQ?

2.The compensation strategy relies on two thresholds, $\theta_1$ and $\theta_2$, to switch between model-level, Transformer-level, and linear-level compensation. How were these thresholds determined for the experiments reported in the paper? How sensitive is the final performance to their values, and would they need to be re-tuned for different model architectures or bit-widths, potentially limiting the "plug-and-play" nature of the OCP?

---

> ### Author Response · Authors · 2025-11-24
>
> Thank you very much for your careful review and insightful comments. Your questions and suggestions are highly constructive and have prompted us to further reflect on and improve the generality and practicality of our approach.
>
> ## **Q1: Regarding the results in Table 2**
>
> We have reported our experimental results faithfully. When we observed anomalous results in Table 2 (highlighted in bold in the original), we conducted thorough checks. We believe these anomalies may be attributed to the interaction between the inherent optimization methods of GPTVQ and AQLM and our additional OCP optimization, which may have amplified the randomness of the dataset. In fact, similar unexpected results have appeared multiple times in our accuracy evaluations. The first instance is the one you pointed out, where perplexity decreases but accuracy also drops. The second case, discussed in Appendix E.3.3, is where we found that after quantization, the perplexity surpassed that of FP16 precision, while the accuracy was lower than anticipated (note that OCP was not involved in this particular instance).
>
> Therefore, we have ruled out the possibility that “the accuracy of OCP may depend on the underlying quantization scheme.” Additionally, when investigating Reviewer 7kTW’s suggestion regarding “the impact of calibration sets on quantization performance,” we found that other studies [1] have reported similar phenomena, i.e., quantization effectiveness is highly sensitive to the choice of calibration set and algorithmic combinations, and there exists a complex relationship between perplexity and accuracy. This further corroborates that the patterns observed in our experiments are not isolated cases.
>
> In summary, we believe that quantization effects are influenced by various factors, including dataset domain characteristics, knowledge biases, and data quality. As such, a single metric is insufficient to fully capture overall model performance. Given the multiple evaluation metrics in our measurements, we consider the effectiveness of OCP to be well supported.
>
> ---
>
> **Table 1: Accuracy of different bit quantization methods on Llama3-8B across various tasks (2-bit results only)**
>
> | Method       | #Bits | Hums. | STEM  | Social | Other | Avg.  | QNLI  | MNLI  | advglu | T.mc1 | T.mc2 |
> |--------------|-------|-------|-------|--------|-------|-------|-------|-------|--------|-------|-------|
> | AQLM         | 2     | 42.23 | 39.63 | 50.02  | 47.84 | 44.67 | 49.75 | 36.41 | 43.36  | 25.58 | 42.40 |
> | AQLM+OCP     | 2     | **40.09** | 41.83 | 51.51  | 50.74 | **45.63** | **49.62** | 38.90 | 43.36  | **25.34** | 42.85 |
> | VPTQ         | 2     | 42.12 | 38.84 | 50.99  | 42.92 | 43.69 | 49.21 | 34.54 | 43.22  | 25.21 | 46.35 |
> | VPTQ+OCP     | 2     | 43.18 | 40.92 | 52.17  | 45.86 | 45.53 | 49.62 | 36.78 | 42.94  | 26.89 | 46.90 |
> | GPTVQ        | 2     | 28.86 | 24.12 | 30.94  | 24.89 | 26.81 | 48.49 | 33.67 | 43.36  | 26.56 | 47.79 |
> | GPTVQ+OCP    | 2     | **24.10** | 28.96 | **30.78** | **24.86** | **26.78** | **48.12** | **33.46** | 43.36  | **26.19** | 47.80 |
> | NuBitQ+OCP   | 2     | 49.48 | 48.61 | 67.47  | 64.81 | 56.77 | 49.66 | 49.60 | 43.77  | 29.63 | 47.58 |
>
> ---
>
> Secondly, when evaluating the combined effects of OCP with other quantization methods, we did not isolate the original optimization techniques from the additional OCP optimization due to the coupling of these methods. As Reviewer 7kTW insightfully pointed out, exploring the interplay and cumulative impact of various optimization techniques is indeed a valuable direction. This investigation fills a gap in understanding how different post-quantization optimization methods interact when combined. To address this, we have conducted targeted experiments, and the results are as follows:
>
> ---
>
> **Table 2: Ablation Study of Various Anomaly Optimization Methods in Non-Uniform Quantization for Llama3-8B**
>
> | Method               | Smooth | Fine-tune | Rotation | OCP   | Time Cost(s) | Memory Cost(kb) | PPL    |
> |----------------------|:------:|:---------:|:--------:|:-----:|:------------:|:---------------:|:------:|
> | before               |        |           |          |       | 0            | 0               | 5.6269 |
> | Uniform Quantization  |   ✓    |           |          |       | 2.75         | 1.188           | 5.6244 |
> |                      |        |     ✓     |          |       | 1622.6       | 0               | 5.6217 |
> |                      |        |           |    ✓     |       | 84.58        | 448.8           | 5.6201 |
> |                      |        |           |          |   ✓   | 45.17        | 114.7           | 5.6014 |
> |                      |   ✓    |           |          |   ✓   | 46.33        | 115.9           | 5.6002 |
> |                      |        |     ✓     |          |   ✓   | 1570.1       | 114.7           | 5.6101 |
> |                      |        |           |    ✓     |   ✓   | 119.05       | 573.4           | 5.5997 |
>
> ---

---

> ### Author Response · Authors · 2025-11-24
> **Regarding the Threshold Selection Strategy**
>
> ## **Q2: Regarding the Threshold Selection Strategy**
>
> How did we determine the threshold combinations in our own experiments? We selected them based on the hyperparameter pruning method described above. However, in practice, only a small number of critical layers significantly affect model performance [2]. Therefore, selecting just the most important layers is generally sufficient—typically, covering the top 10% of layers is enough. Of course, as long as the compression ratio permits, we can apply compensation at a finer granularity.
>
> How sensitive is performance to the threshold values? In multiple experiments, we fine-tuned the values around the chosen thresholds. The results show that the overall performance is reasonably robust to threshold selection; that is, small variations in the threshold within a reasonable range do not significantly impact the final model performance.
>
> ---
>
> **Table 1: Threshold Sensitivity Experiment Results**
>
> | Threshold Selection | Model Level | Transformer Level | Linear Level | ppl (wikitext-2) | Throughput (tokens/s) | Memory Usage (MB) |
> |---------------------|-------------|-------------------|--------------|------------------|-----------------------|-------------------|
> | 0                   | 32          | 0                 | 0            | 6.5368           | 30.93                 | 8759.38           |
> | 1                   | 24          | 8                 | 0            | 6.5498           | 31.15                 | 8760.24           |
> | 2                   | 16          | 16                | 0            | 6.5429           | 31.58                 | 8760.50           |
> | 3                   | 8           | 24                | 0            | 6.5237           | 31.88                 | 8760.73           |
> | 4                   | 0           | 24                | 8            | 6.4759           | 32.35                 | 8760.93           |
> | 5                   | 0           | 16                | 16           | 6.3734           | 32.62                 | 8761.22           |
> | 6                   | 0           | 8                 | 24           | 6.2833           | 32.72                 | 8761.47           |
> | 7                   | 0           | 0                 | 32           | 6.2761           | 32.83                 | 8761.77           |
>
> ---
>
> Is redesign necessary? For transformers, it is generally not required. Although factors such as model architecture (e.g., number of parameters and layers) and quantization bit-width can influence the distribution of outliers—and thus may necessitate some adjustment of thresholds to suit new scenarios—in practical settings we typically perform layer-wise compensation immediately after quantization based on available memory. This naturally forms a multi-objective optimization problem alongside other hyperparameters, where we need to determine the optimal strategy for each layer and select appropriate thresholds so that the overall compression meets the desired bit budget while minimizing performance degradation.
>
> Conventionally, this would require considering all parameters and constraints simultaneously. However, our primary goal has always been to achieve the most suitable quantization results at minimal cost. Fortunately, through our hyperparameter ablation studies, we found that the parameters *r* and *d* have the most significant impact. This enables us to perform hyperparameter pruning, focusing primarily on *r*, *d*, and the compression bit constraints to find the optimal combination.
>
> We treat compensation as a sequential process; under the bit budget constraint, the threshold combinations are refined to the finest granularity possible to maximize compensation effectiveness.
>
> ---
>
> Once again, we sincerely thank you for your meticulous and insightful review. Your feedback has not only helped us identify and improve the shortcomings of our current work but also inspired us to enhance its generality, usability, and scientific rigor. We look forward to addressing your concerns comprehensively in the revised manuscript and advancing research progress in this field.
>
> ---
>
> ### **References**
>
> [1] M. Williams and N. Aletras, "On the Impact of Calibration Data in Post-training Quantization and Pruning," ACL 2023.
> [2] V. Malinovskii et al., "Pushing the Limits of Large Language Model Quantization via the Linearity Theorem," 2024.

---

### Official Review · Reviewer_343g · 2025-11-01

**Soundness:** 2
**Presentation:** 3
**Contribution:** 2
**Rating:** 4
**Confidence:** 4

**Summary:**

This paper proposes a **fine-grained, multi-codebook, multi-vector quantization** strategy that adapts bit-width and codebook design on a per-layer basis. In addition, an **outlier-based compensation** mechanism is introduced to further improve the overall quantization performance.

**Strengths:**

1. The proposed outlier compensation plugin effectively enhances the performance of the quantized model.
2. The evaluation covers multiple types of LLMs and diverse downstream tasks, demonstrating general applicability.

**Weaknesses:**

1. The discussion in the appendix omits some important state-of-the-art baselines, such as **DuQuant** and **ResQ**.
2. Certain results are not state-of-the-art but are marked as “best” in Table 2, which may cause confusion.
3. The proposed non-uniform quantization may introduce additional inference overhead, which is not clearly discussed.
4. The abbreviation **OCP** is introduced in the Introduction without prior definition or explanation.

**Questions:**

1. Could you include additional baselines such as **DuQuant** and **ResQ** in **Table 4** for fairer comparison?
2. What is the additional computational or inference cost introduced by **NuBitQ**?
3. Could you provide end-to-end measurements of speedup and memory usage to validate the claimed efficiency?

---

> ### Author Response · Authors · 2025-11-24
> **W1 and W2**
>
> Thank you for your thorough and thoughtful review！ Your guidance has been critical for us to refine our work from both the perspectives of scholarly rigor and deployment value, thereby enhancing the overall quality of our manuscript.
>
> ## **W1 state-of-the-art baselines**
>
> We have incorporated comparisons with the latest baselines you mentioned, such as DuQuant and ResQ, into the appendix and have also included discussions of these methods in the related work section. We also noticed that these recent baseline methods address quantization outliers via two popular solutions: smoothing and matrix rotation. Following the suggestion from Reviewer 7kTW, we adopted the rotation and decomposition concepts from DuQuant and ResQ to conduct an ablation study on our outlier optimization component. The results demonstrate that, in terms of method combinations, Smooth and Rotation combined with OCP yield cumulative improvements, whereas Fine-tune causes performance degradation. The fundamental reason is that OCP and similar outlier-focused methods complement global optimizations like Smooth and Rotation, which target non-outlier channels, thereby achieving synergistic enhancement. In contrast, Fine-tune’s global optimization strategy conflicts with the targeted outlier corrections, often causing mutual interference or degradation that ultimately harms overall performance.
>
> | Method                | Smooth | Fine-tune | Rotation | OCP   | Time Cost(s) | Memory Cost(kb) | PPL    |
> |-----------------------|:------:|:---------:|:--------:|:-----:|:---------:|:-----------:|:-------|
> | **before  optimization**            |        |           |          |       | 0         | 0           | 5.6269  |
> |  | ✓      |           |          |       | 2.75      | 1.188       | 5.6244  |
> |                       |        | ✓         |          |       | 1622.6    | 0           | 5.6217  |
> |                       |        |           | ✓        |       | 84.58     | 448.8       | 5.6201  |
> |                       |        |           |          | ✓     | 45.17     | 114.7       | 5.6014  |
> |                       | ✓      |           |          | ✓     | 46.33     | 115.9       | 5.6002  |
> |                       |      | ✓         |          | ✓     | 1570.1    | 114.7       | 5.6101  |
> |                       |    |       | ✓        | ✓     | 119.05    | 573.4       | 5.5997  |
>
> ---
>
> ## **W2  Unexpected results**
>
> Following your suggestion, we have revised the wording in the final manuscript to make it more precise and rigorous. It should be noted that our method indeed achieves the best overall performance among the compared approaches in Table 2.You may have noticed that accuracy declines on certain tasks, as indicated by the bolded entries in the table. Regarding these anomalous results, we conducted a careful investigation and believe that they mainly stem from the interaction between the optimizations of GPTVQ and AQLM methods themselves and our proposed OCP optimization, which amplifies dataset-specific randomness in performance. For further reference, the issue of optimization method stacking can be seen in the tables provided in W1, while the dataset randomness is discussed in detail in Appendix E.3.3.
>
> | Method     | #Bits | Knowledge (%) ↑                       |       |       |       |       |       | Inference (%) ↑ |       | Trustworthiness (%) ↑     |      |      |
> |------------|-------|-------------------------------------|-------|-------|-------|-------|-------|-----------------|-------|---------------------------|------|------|
> |            |       | Hums.                               | STEM  | Social| Other | Avg.  | QNLI  | MNLI            | advglu| T.mc1                     | T.mc2|      |
> | AQLM       | 2     | 42.23                               | 39.63 | 50.02 | 47.84 | 44.67 | 49.75 | 36.41           | 43.36 | 25.58                     | 42.40|      |
> | AQLM+OCP   | 2     | **40.09**                          | 41.83 | 51.51 | 50.74 | **45.63** | **49.62** | 38.90           | 43.36 | **25.34**                 | 42.85|      |
> | VPTQ       | 2     | 42.12                               | 38.84 | 50.99 | 42.92 | 43.69 | 49.21 | 34.54           | 43.22 | 25.21                     | 46.35|      |
> | VPTQ+OCP   | 2     | 43.18                               | 40.92 | 52.17 | 45.86 | 45.53 | 49.62 | 36.78           | 42.94 | 26.89                     | 46.90|      |
> | GPTVQ      | 2     | 28.86                               | 24.12 | 30.94 | 24.89 | 26.81 | 48.49 | 33.67           | 43.36 | 26.56                     | 47.79|      |
> | GPTVQ+OCP  | 2     | **24.10**                          | 28.96 | **30.78** | **24.86** | **26.78** | **48.12** | **33.46**           | 43.36 | **26.19**                 | 47.80|      |
> | NuBitQ+OCP | 2     | 49.48                               | 48.61 | 67.47 | 64.81 | 56.77 | 49.66 | 49.60           | 43.77 | 29.63                     | 47.58|      |

---

> ### Author Response · Authors · 2025-11-24
>
> ## **W3 additional inference overhead**
> Regarding the additional inference overhead introduced by our method:
> - **Increased lookup operations:** Unlike uniform quantization, which recovers weights via simple arithmetic operations, non-uniform quantization requires accessing lookup tables each time to retrieve quantized weight values.
> - **Extra computation for outlier channel processing:** Our approach applies specialized optimizations (e.g., the Outlier Compensation Plugin, OCP) targeting outlier channels, which inevitably incur some additional compute and parameter storage during inference.
>
> However, these extra costs are generally manageable for the following reasons:
> - Quantization overall significantly reduces model memory footprint and memory bandwidth demands. This compression effectively alleviates data transfer bottlenecks in memory-intensive inference scenarios, thus improving overall throughput.
> - With the advent of highly optimized quantized inference engines such as ExLlamaV2 and Marlin, lookup operations and codebook-integrated matrix multiplications can be efficiently executed at the low-level (e.g., GEMM libraries or GPU instruction sets). Techniques like offline matrix reconstruction, vectorized lookup tables, and streamed decomposition computations effectively overcome the traditional performance bottlenecks associated with table lookups. As a result, non-uniform quantization methods can achieve inference speeds comparable to or even surpassing uniform quantization in practical deployment.
> - Our outlier channel optimization typically targets only a very small subset of weights, so the overall computational overhead is minimal. The incremental cost of these localized corrections is far outweighed by the bandwidth and storage savings enabled by quantization.
>
> For actual inference performance, please refer to our response in Q3.
>
> ---
>
> ## **W4 OCP  definition**
> In fact, we have already defined the Outlier Compensation Plugin (OCP) in the abstract section.
>
> ---
>
> ## **Q1**
> Same as W1.
>
> ---
>
> ## **Q2**
> Same as W2.
>
>
> ###  **Q3  speedup and memory usage**
> Similar to Reviewer pUW6, you are also very concerned about the performance and efficiency of our method. In response to the various reviewers’ requests for energy efficiency comparisons, we designed experiments that include acceleration and memory usage metrics according to your requirements. These are our final results（2048 tokens are pre-filled, and 128 tokens are decoded at each step）.
> ### Table: End-to-End Evaluation Metrics
>
> | Method   | Lat. | Thpt. | Mem. | Lat. | Thpt. | Mem. | Lat. | Thpt. | Mem. |
> |----------|------|-------|------|------|-------|------|------|-------|------|
> |          | **LLaMA3-8B on Ada** |       |      | **LLaMA3-8B on Ampere** |       |      | **LLaMA3-70B on Ampere** |       |      |
> | INT8     | 179.00 | 0.72  | 9721  | 5.26  | 25.35 | 9227  | 22.00  | 5.82  | 77816  |
> | FP16     | 363.62 | 0.35  | 16587 | 13.22 | 9.68  | 16587 | 44.77  | 2.86  | 143756 |
> | AQLM     | 609.93 | 0.21  | 7400  | 194.13| 0.66  | 7054  | 688.34 | 0.19  | 62804  |
> | VPTQ     | 62.04  | 2.06  | 17416 | 4.59  | 27.88 | 17473 | 16.94  | 7.53  | 143600 |
> | GPTVQ    | 54.51  | 2.35  | 19476 | 4.03  | 31.79 | 19476 | 14.90  | 8.58  | 157356 |
> | ours     | 55.24  | 2.32  | 8974  | 4.20  | 30.50 | 8766  | 15.94  | 8.03  | 72916  |
>
> - **Lat.**: Latency  (s)
> - **Thpt.**: Throughput(token/s)
> - **Mem.**: Memory Usage (mb)
>
> Thank you once again for your valuable feedback and professional suggestions. We have made corresponding additions and revisions based on your comments, and we hope these improvements will make the paper more complete and a significant contribution to the conference. If you have any further suggestions for revisions, please kindly let us know; otherwise, we would greatly appreciate your support in raising the evaluation score for our paper.

---

> ### Author Response · Authors · 2025-11-27
>
> Dear Reviewer 343g,
>
> I hope this message finds you well! As the discussion period is coming to a close, we warmly welcome any additional valuable feedback you may have. Your insights are incredibly important to us, and we truly appreciate your thoughtful guidance and support throughout.
>
> Thank you again for your time and help!

---

### Official Review · Reviewer_7kTW · 2025-11-01

**Soundness:** 3
**Presentation:** 3
**Contribution:** 2
**Rating:** 4
**Confidence:** 2

**Summary:**

The paper proposes NuBitQ (a vector-quantization-style, layer-wise, residual codebook design) and OCP (a three-level compensation guided by a novel block-level outlier score). It achieves near-lossless 4-bit and strong 2–3-bit results on several LLMs, without Hessian or fine-tuning.

**Strengths:**

1. Identifies a different outlier pattern for non-uniform quantization (block-level), and formalizes a usable score.

2. Practical PTQ: no Hessian, no fine-tuning; clear knobs (𝑟, 𝑐, 𝑑, 𝑔) with an explicit compression formula.

3. Consistent empirical gains at 4/3/2 bits across model families; ablations on hyper-params and compensation granularity are informative.

**Weaknesses:**

1. While the method avoids fine-tuning/Hessian, it still requires a nontrivial search over (r,c,d,g) and threshold(s) to balance accuracy vs. memory/latency. The paper shows trends, but lacks a budget-constrained, step-by-step selection procedure (e.g., how to pick (r,d) first, then threshold, under a fixed memory/latency cap). This limits immediate reproducibility in large-scale deployments.

2. The paper does not discuss systems constraints: current vendor fast paths primarily target uniform INT/FP formats; non-uniform (codebook/LUT) typically needs custom kernels or on-the-fly dequant, making throughput sensitive to group size, batch size, and cache behavior. A brief analysis of runtime/throughput vs. baseline INT8/FP8 would clarify practicality.

3. Prior work shows that rotation/affine transforms can mitigate outliers before quantization. The paper compares methods side-by-side, but does not systematically evaluate combinations (e.g., rotation → non-uniform weight quant → OCP). A combined study could reveal whether gains are additive or saturating, and where each component contributes.

**Questions:**

1. How stable is the outlier score ranking under domain shift and smaller calibration sets?

2. Can OCP selection be cast as a budgeted optimization to auto-derive Pareto frontiers?

3. What is the end-to-end latency/throughput impact on 70B-scale serving?

---

> ### Author Response · Authors · 2025-11-24
> **for w1 and w2**
>
> **Thank you for your detailed and insightful feedback!**
> We appreciate your focus on the method's practicality and system performance, which will help us improve automated tuning, robustness evaluation, and large-scale deployment. In response to your comments, we will make the following updates and improvements:
>
> ---
>
> ## **W1 Hyperparameter pick**
>
> The same as Q2
>
> ---
>
> ## **W2 systems constraints**
>
> Indeed, as you pointed out, a common drawback of non-uniform quantization methods is that although they often achieve significantly higher accuracy than uniform quantization, their actual computational performance can be worse due to the overhead of extensive lookup table operations, which add extra cost compared to straightforward calculations. However, since quantization reduces memory traffic and inference is typically limited by memory bandwidth—especially on edge devices and GPUs with limited VRAM—moving large amounts of weight and activation data becomes a bottleneck. Overall, quantization can still provide acceleration.
>
> Recently, many low-level optimization techniques for quantization have emerged, such as ExLlamaV2 and Marlin. Marlin, for example, combines lookup tables with GEMM operations [1], addressing memory bandwidth limits and challenges from low-bit quantization by offline reconstruction of quantized weight matrices, vectorized lookup tables, and streaming K-work decomposition, thereby accelerating inference.
>
> Here, we provide a performance comparison between non-uniform 8-bit lookup table quantization and uniform INT8/FP8 quantization (a relatively fast baseline):
>
> **Table 1. End-to-End Evaluation Metrics. Prefill max input length is 2048 to simulate real context; decoding generates 128 tokens per step. Lat.: Latency (s), Thpt.: Throughput (token/s), Mem.: Peak Memory (MB).**
>
> | Method | Lat. | Thpt. | Mem. | Lat. | Thpt. | Mem. |
> |---|---|---|---|---|---|---|
> |      | **LLaMA3-8B** |  |  | **LLaMA3-70B** |  |  |
> | INT8 | 5.26 | 25.35 | 9227 | 22.00 | 5.82 | 77816 |
> | ours | 4.20 | 30.50 | 8766 | 15.94 | 8.03 | 72916 |
>
> In addition, addressing the feedback from Reviewers pUW6 and 343g, we have compared the performance of various baseline methods against FP16, and the final experimental results will be consolidated together.
>
> ---

---

> ### Author Response · Authors · 2025-11-24
> **for w3**
>
> ## **W3 evaluate combinations**
>
> Rotation and similar transformations are optimized based on outlier channels identified in uniform quantization. Our experiments indicate that, in non-uniform quantization, the quantization errors do not exhibit a clear channel-wise distribution. Prior work [2] also shows that quantization error is strongly related to the clipping method used. Therefore, we only provide a brief discussion on rotation in this context. However, like you, we are very interested in the combined effects of various post-quantization optimization methods such as fine-tuning and OCP.
>
> To investigate this, we designed a series of ablation experiments on post-quantization optimization methods. The experimental setups and specific methods are as follows:
>
> - **Smooth**: Following the smooth setting, we apply scaling factors to smooth the activated outlier channels.
> - **Fine-tune**: We adopt the fine-tuning procedure as described in AQLM.
> - **Rotation**: We use the dequant [4] and ResQ [5] rotation method with parameters set to block size = 128 and rotation steps = 256.
> - **OCP**: We apply transformer-level compensation for outlier channels.
>
> **Table 2. Ablation Study of Various Anomaly Optimization Methods in Non-Uniform Quantization for Llama3-8B**
>
> | Method    | Smooth | Fine-tune | Rotation | OCP | Time Cost (s) | Memory Cost (kb) | PPL    |
> |-----------|--------|-----------|----------|-----|---------------|------------------|--------|
> | before    |        |           |          |     | 0             | 0                | 5.6269 |
> |           | ✓      |           |          |     | 2.75          | 1.188            | 5.6244 |
> |           |        | ✓         |          |     | 1622.6        | 0                | 5.6217 |
> |           |        |           | ✓        |     | 84.58         | 448.8            | 5.6201 |
> |           |        |           |          | ✓   | 45.17         | 114.7            | 5.6014 |
> |           | ✓      |           |          | ✓   | 46.33         | 115.9            | 5.6002 |
> |           |        | ✓         |          | ✓   | 1570.1        | 114.7            | 5.6101 |
> |           |        |           | ✓        | ✓   | 119.05        | 573.4            | 5.5997 |
>
> Experimental results show that Smooth and Rotation adjustments have limited effectiveness for non-uniform quantization, as they primarily optimize anomalous channels. Among individual methods, OCP achieves the best performance, striking a balance between time, memory usage, and the lowest perplexity.
>
> When combining methods, both Smooth and Rotation exhibit additive improvements alongside OCP, whereas Fine-tune leads to a degradation in performance. The underlying reason is that OCP and similar anomaly-focused methods complement global non-anomalous optimizations like Smooth and Rotation, enabling synergistic enhancements. In contrast, Fine-tune’s global optimization conflicts with the targeted anomaly corrections, often causing mutual interference or dilution of effects, resulting in worse overall performance.

---

> ### Author Response · Authors · 2025-11-24
>
> ## **Q1  calibration sets**
>
> What a coincidence! When we first learned about layer-wise fine-tuning as a post-quantization optimization method, we were very curious about how the choice of calibration dataset affects quantization performance. At that time, we conducted experiments on the impact of different calibration sets on the quantization of llama3-8B. Now, our quantization compensation strategy also incorporates experiments using various calibration datasets.
>
> |                | **Wikitext2** | **Ptb** | **Red prajama** |
> |----------------|---------------|---------|-----------------|
> | Fine-tune      | 7.2771        | 7.3756  | 7.6021          |
> | OCP            | 6.4228        | 6.4467  | 6.4329          |
>
> We observe that, whether using fine-tuning or OCP, quantization performance is consistently better when evaluated on the calibration set compared to other datasets. This is consistent with the findings in [2].
>
> However, when applying the OCP method, we notice that the differences between calibration sets become much smaller. Specifically, this is because fine-tuning adjusts the entire codebook based on the dataset, aiming to minimize overall loss across the dataset; therefore, the quality and diversity of the dataset have a greater impact on fine-tuning. In contrast, OCP compensates specifically for anomalies, and only anomalous instances within the dataset are addressed. As a result, the influence of the dataset's quality and coverage is more pronounced for fine-tuning than for OCP.
>
> ---
>
> ## **Q2 parameter search**
> First, thank you for recognizing that our method indeed avoids the use of fine-tuning and Hessian computations. In fact, we
> discuss the quantization hyperparameters in Appendix B. Our algorithm optimizes the selection of candidate quantization parameter combinations based on performance metrics under a fixed model size constraint, achieving an effective balance between performance and model compression. This reflects an approximate Pareto-optimal parameter search strategy.
>
> Specifically, for parameter search, we first generate a candidate set of multiple parameter configurations. Then, through constrained combinatorial optimization—similar to a knapsack problem—we efficiently search for the optimal combination within this candidate set. This approach avoids exhaustive brute-force exploration of the entire parameter space, offering both good search efficiency and practicality. In practice, the most time-consuming part is generating the multi-parameter candidate set. Based on our hyperparameter ablation studies, where accuracy is relatively insensitive to changes in *g* and *c*, we prune the search space by focusing on diversity in *r* and *d*. On Llama3-8B, collecting parameter data takes about 200 seconds per layer, which is required by most optimization methods. The table below compares resource costs for quantizing a single layer, highlighting the efficiency of our method.
>
> Regarding OCP’s threshold search, it is inherently automatic: after obtaining the available system memory, the highest-precision compensation is enabled within the allowed memory limits.
>
> | **method**     | **Time Cost(s)** | **Memory Cost(kb)** |
> |--------------|--------------:|---------------:|
> | Smooth       |          2.75 |          1.188 |
> | Fine-tune    |       1622.6  |          0     |
> | Rotation     |         84.58 |         448.8  |
> | OCP          |         45.17 |         114.7  |
>
> ---
>
> ## **Q3 end-to-end performance of the 70B model**
>
> Regarding your concern about the end-to-end performance of the 70B model, we have conducted experiments. Below are the results for LLama3-70B. Experimental setup: 2048 tokens are pre-filled, and decoding is performed in steps of 128 tokens each.
>
> | Architecture | Method  | Latency (s) | Throughput (tokens/s) | Peak Memory (MB) |
> |--------------|---------|-------------|----------------------|------------------|
> | **Ampere**   | INT8    | 22.00       | 5.82                 | 77816            |
> |              | FP16    | 44.77       | 2.86                 | 143756           |
> |              | AQLM    | 688.34      | 0.19                 | 62804            |
> |              | VPTQ    | 16.94       | 7.53                 | 143600           |
> |              | GPTVQ   | 14.90       | 8.58                 | 157356           |
> |              | ours    | 15.94       | 8.03                 | 72916            |
>
> ---
>
> Thank you again for your careful review and valuable feedback. Your comments led us to a deeper analysis of our method’s practical deployment and system performance. This helped us better understand and improve quantization’s real-world behavior and address prior application gaps.

---

> ### Author Response · Authors · 2025-11-27
> **References**
>
> ### References
>
> [1] Han Guo et al., *Fast matrix multiplications for lookup table-quantized LLMs*, EMNLP 2024.
>
> [2] Zhuocheng Gong et al., *What makes quantization for large language model hard? An empirical study from the lens of perturbation*, AAAI 2024.
>
> [3] Miles Williams and Nikolaos Aletras, *On the Impact of Calibration Data in Post-training Quantization and Pruning*, ACL 2023.
>
> [4] Haokun Lin et al., *DuQuant: Distributing Outliers via Dual Transformation Makes Stronger Quantized LLMs*, NeurIPS 2024.
>
> [5] Utkarsh Saxena et al., *ResQ: Mixed-Precision Quantization of Large Language Models with Low-Rank Residuals*, 2024.

---

### Official Review · Reviewer_pUW6 · 2025-11-01

**Soundness:** 3
**Presentation:** 3
**Contribution:** 3
**Rating:** 8
**Confidence:** 4

**Summary:**

This paper proposes NuBitQ, a flexible non-uniform quantization framework for large language models (LLMs), along with a plug-and-play Outlier Compensation Plug-in (OCP) to address outlier-induced performance degradation in low-bit settings. Extensive experiments showing near-lossless 4-bit performance and strong results at 2–3 bits across multiple LLMs.

**Strengths:**

- Introduces a novel outlier score that integrates both local sensitivity and global propagation effects, which is more tailored to non-uniform quantization than prior Hessian-based methods.

- The OCP module is a lightweight, tuning-free compensation mechanism that is also applicable to other quantization methods.

- Supports arbitrary bit-widths and layer-wise customization, offering more flexibility than fixed-bit approaches.

- Comprehensive evaluations on multiple models (LLaMA, Qwen, Gemma) and tasks (perplexity, MMLU, QNLI, MNLI, etc.) demonstrate consistent improvements over SOTA methods.

**Weaknesses:**

- Lack of Hardware-Aware Latency Evaluation : Although Table 3(b) compares Ada vs. Ampere architectures, there is no end-to-end latency or throughput comparison against FP16 or other quantization methods.

- Lack of Direct Validation:The experimental results do not directly validate these specific theoretical claims.

**Questions:**

- Can you provide the end-to-end latency or throughput comparison against FP16 or other quantization methods.

- Could the authors present quantitative or visual evidence (e.g., density plots or clustering metrics) supporting this claim?

---

> ### Author Response · Authors · 2025-11-24
>
> Thank you for your recognition of our work and your valuable suggestions! You not only focus on the innovation of our method, but the issues you raised are also crucial for improving our research. In response to your feedback, we will make the following improvements:
>
> ## **Q1 Supplement: Comparison of End-to-End Latency and Throughput**
>
> We highly appreciate your emphasis on practical performance and rigorous validation, which is invaluable to our research. We will conduct additional experiments comparing end-to-end latency and throughput of FP16 and mainstream non-uniform quantization methods under the same hardware and environment, and present the results clearly in a table.
>
> ---
>
> **Table 1: End-to-End Evaluation Metrics — Latency (s), Throughput (tokens/s), Peak Memory (MB)**
>
> | Architecture | Method  | Latency (s) | Throughput (tokens/s) | Peak Memory (MB) |
> |--------------|---------|-------------|----------------------|------------------|
> | **Ampere**   | INT8    | 5.26        | 25.35                | 9227             |
> |              | FP16    | 13.22       | 9.68                 | 16587            |
> |              | AQLM    | 194.13      | 0.66                 | 7054             |
> |              | VPTQ    | 4.59        | 27.88                | 17473            |
> |              | GPTVQ   | 4.03        | 31.79                | 19476            |
> |              | ours    | 4.20        | 30.50                | 8766             |
> | **Ada**      | INT8    | 179.00      | 0.72                 | 9721             |
> |              | FP16    | 363.62      | 0.35                 | 16587            |
> |              | AQLM    | 609.93      | 0.21                 | 7400             |
> |              | VPTQ    | 62.04       | 2.06                 | 17416            |
> |              | GPTVQ   | 54.51       | 2.35                 | 19476            |
> |              | ours    | 55.24       | 2.32                 | 8974             |
>
> ---
>
> Based on the experimental results, non-uniform quantization methods—especially VPTQ, GPTVQ, and our proposed approach—demonstrate significant advantages in latency and throughput by alleviating memory bandwidth bottlenecks common in inference scenarios, thus accelerating performance. Our “ours” scheme shows consistent performance across both architectures, achieving low latency, high throughput, and moderate memory usage, which highlights its practical value and broad applicability.
>
> Additionally, some data may raise questions—for example, why VPTQ and GPTVQ do not show the expected reduction in peak memory usage. This is because, although their static parameter memory achieves 2-bit compression, dynamic computations require dequantization to match activation precision, resulting in peak memory usage similar to FP16. Regarding the high latency of AQLM, it is due to its large codebook size; the model employs a massive codebook (with each codebook entry being 2^16), which increases computational overhead.

---

> ### Author Response · Authors · 2025-11-24
> **Response to Q2**
>
> ## **Response to Q2: Add quantitative and visual validation**
>
> Thank you for your valuable feedback regarding the quantitative and visual validation of our theory that *“non-uniform quantization exhibits outliers distinct from those in uniform quantization”*.
>
> Prior work [1] defines the quantization error for uniform quantization and highlights the presence of so-called "outlier channels" in large language models (LLMs), where certain feature dimensions have values significantly higher than others. In our paper, Figures 1 and 5 demonstrate that the output quantization error from non-uniform quantization does not follow a channel-wise pattern.
>
> To further support our analysis, we present clustering metrics for Llama3-8B at the block level:
>
> - **Silhouette Coefficient:** Measures how well clusters are separated and how compact they are; ranges from -1 to 1, with values closer to 1 indicating better clustering.
> - **Mean Intra-cluster Distance:** The average distance between points within the same cluster; smaller values indicate tighter clusters.
> - **Mean Inter-cluster Distance:** The average distance between points in different clusters; larger values indicate well-separated clusters.
>
> **Table 1. Statistics of Clustering Metrics for our Quantization in Each Block**
>
> | Block Index | Silhouette Coefficient | Mean Intra-cluster Distance | Mean Inter-cluster Distance |
> |-------------|-----------------------|-----------------------------|-----------------------------|
> | 0  | 0.61380 | 0.3007 | 0.1541 |
> | 1  | 0.61225 | 0.2976 | 0.1663 |
> | 2  | 0.61376 | 0.2873 | 0.1525 |
> | 3  | 0.61206 | 0.2998 | 0.1819 |
> | 4  | 0.61149 | 0.2857 | 0.1465 |
> | 5  | 0.61216 | 0.2751 | 0.1482 |
> | 6  | 0.61343 | 0.2747 | 0.1564 |
> | 7  | 0.61370 | 0.2698 | 0.1436 |
> | 8  | 0.61154 | 0.3046 | 0.1601 |
> | 9  | 0.61278 | 0.2809 | 0.1576 |
> | 10 | 0.61440 | 0.2849 | 0.1475 |
> | 11 | 0.61202 | 0.2840 | 0.1482 |
> | 12 | 0.61346 | 0.2766 | 0.1462 |
> | 13 | 0.61207 | 0.2652 | 0.1402 |
> | 14 | 0.61255 | 0.3142 | 0.1720 |
> | 15 | 0.61245 | 0.2606 | 0.1384 |
> | 16 | 0.61435 | 0.2938 | 0.1524 |
> | 17 | 0.61186 | 0.2806 | 0.1533 |
> | 18 | 0.61238 | 0.2975 | 0.1520 |
> | 19 | 0.61220 | 0.2845 | 0.1541 |
> | 20 | 0.61545 | 0.2904 | 0.1561 |
> | 21 | 0.61216 | 0.2957 | 0.1562 |
> | 22 | 0.61251 | 0.2911 | 0.1590 |
> | 23 | 0.61750 | 0.2850 | 0.1427 |
> | 24 | 0.62029 | 0.3395 | 0.1881 |
> | 25 | 0.61267 | 0.3018 | 0.1574 |
> | 26 | 0.61350 | 0.2755 | 0.1658 |
> | 27 | 0.61146 | 0.2702 | 0.1422 |
> | 28 | 0.61236 | 0.3013 | 0.1588 |
> | 29 | 0.61056 | 0.2715 | 0.1434 |
> | 30 | 0.61128 | 0.2848 | 0.1529 |
> | 31 | 0.61324 | 0.3221 | 0.1724 |
> | **Average** | **0.61305** | **0.2890** | **0.1552** |
>
> Our results show that the clustering of non-uniform quantization errors does not follow a distinct channel-wise pattern as observed in uniform quantization, consistent with our visual observations.
>
> ---
>
> **How Non-uniform Quantization Errors Develop into Outliers**
>
> We define *outliers* as errors that significantly impact accuracy. For example, although the overall error within a block is small, certain vectors inside the block can have large errors.
>
> Suppose \( d = 4 \), and after replacing four vectors with codebook entries, the quantization errors are:
>
> $$
> \begin{bmatrix}
> 0.3118336 & 0.0368802 & 0.02044559 & 0.09176590
> \end{bmatrix}
> $$
> $$
> \begin{bmatrix}
> 0.5819435 & 1.1185487 & 0.07737651 & 0.06973236
> \end{bmatrix}
> $$
> $$
> \begin{bmatrix}
> 0.0118336 & 0.0300486 & 0.01504153 & 0.27175903
> \end{bmatrix}
> $$
> $$
> \begin{bmatrix}
> 0.3368979 & 0.0118548 & 1.10819237 & 0.00512973
> \end{bmatrix}
> $$
>
>
>
>
> The activation vectors are:
>
> $$
> \begin{bmatrix}
> 0.4675 \\
> 0.8790 \\
> 0.6623 \\
> 0.8743
> \end{bmatrix}
> $$
>
> Resulting in output errors of:
>
>
> \begin{bmatrix}
> 0.2719\\
> 1.3674 \\
> 0.2794 \\
> 0.9063
> \end{bmatrix}
>
>
> Although the average error per dimension is similar and the overall block error is low due to clustering, some vectors incur large deviations to fit their assigned codebook entries, causing amplified output errors. Furthermore, these scattered errors can be magnified by activations and inter-layer propagation, evolving into significant outliers.
>
> We sincerely appreciate your valuable and insightful feedback, which helped us improve both the hardware implementation and theoretical analysis of our method, thereby enhancing the quality of the paper. Should there be any remaining issues, we welcome your continued guidance and will strive to address them diligently.
>
> ### **Reference**
> [1] Nrusimha, Aniruddha, et al. "Mitigating the Impact of Outlier Channels for Language Model Quantization with Activation Regularization." (2024).

---

### Author Response · Authors · 2025-11-30
**Review Summary and Author Responses**

**Dear Chairs and Reviewers,**


### Thank you all for your time and valuable feedback on our submission. Below we summarize the main strengths and weaknesses identified, along with our replies.
---

## **Strengths**

1. **Novel Scenario:** We acknowledge the unique distribution and compensation approach of outliers in the non-uniform quantization setting. {pUW6, R323, 7kTW, jNs7}
2. **Comprehensive Evaluation:** The experiments are thorough and convincing, with extensive validation across multiple models and tasks. {pUW6, 7kTW, 343g, R323, jNs7}
3. **Practical Method:** The method does not require Hessian matrix computation and is clearly explained. {pUW6, 7kTW}
4. **Flexible and Generalizable:** The outlier compensation plugin is designed as a modular, plug-and-play component. {pUW6, R323}
---

## **Weaknesses**

1. **Lack of End-to-End Evaluation of Inference Acceleration {pUW6, 7kTW, 343g, jNs7}**
   **Response:** We conducted multi-metric experiments across different models and architectures, including FP16 and INT8 methods, measuring latency, throughput, and peak memory usage. The results show that our method performs stably on both architectures, achieving low latency, high throughput, and moderate memory usage, demonstrating its practical value and broad applicability.

2. **Parameter Search {7kTW, R323}**
   **Response:** Building upon the hyperparameter ablation studies in the main paper and the hyperparameter search algorithm in Appendix B, we further supplemented the search procedure and timing details. We also added experiments comparing the cost and threshold sensitivity of OCP with other methods. These results indicate that our search method is efficient and approximately Pareto-optimal.

3. **Concerns About Experimental Results {343g, R323}**
   Regarding perplexity reduction accompanied by occasional accuracy drops during evaluation:
   **Response:** We found that this phenomenon is not unique to our work. Combining the experiments in Appendix E.3.3 and our post-quantization optimization method ensemble, as well as reviewing related literature, we conclude that the main cause lies in the interaction between the optimizations of GPTVQ, AQLM, and our proposed OCP, which amplifies dataset-specific performance variability.

4. **Questions on the OCP Method {7kTW, jNs7}**
   - Reviewer 7kTW requested evaluation of the combined effect of Quantization optimization method.
   - Reviewer jNs7 questioned the novelty of OCP, suggesting it is similar to AQLM’s multi-stage optimization.

   **Response:**
   - To address 7kTW’s comment, we supplemented performance and cost comparisons among various quantization optimization methods, concluding that OCP effectively balances performance and synergizes with Smooth and Rotation to improve results.
   - To address jNs7’s concern, we provided comparative experiments in Appendix E.3.2 and additional quantization optimization method tests, which have been acknowledged by the reviewer.

5. **Others:**
   - pUW6 requested visual evidence supporting the “Title” claim — **done**
   - 7kTW inquired about the stability of outlier scoring under domain shifts and small calibration sets. **Based on recent research and comparative experiments, we conclude that calibration set choice influences results, but fine-tuning is more sensitive to data quality, causing significant performance differences across calibration sets; OCP focuses solely on outlier compensation, significantly reducing variability between calibration sets.**
   - 343g suggested adding  DuQuant and ResQ in the appendix discussion —**done**
   - jNs7 pointed out insufficient information in figures and tables — **we have provided specific examples to improve clarity.**
   - jNs7 questioned the outlier scoring (mainly regarding nonlinear effects). **Combining recent research and experimental data comparing inclusion and exclusion of nonlinear activations, we clarify that linearity is an observed phenomenon and our method already incorporates nonlinear factors.**

---

## **Acknowledgments**

**Regarding the practicality of the method**，We thank {pUW6, 7kTW, 343g, jNs7} for their attention to practical inference acceleration, prompting us to supplement end-to-end inference results. We appreciate 7kTW and R323’s curiosity about parameter search, allowing us to provide further clarifications.

Special thanks to {343g, R323} for their **careful review of our experimental data**, enabling us to discover interesting insights from unexpected results.

We also sincerely thank 7kTW for proposing the combination effects of quantization optimization methods, **filling an important gap in post-quantization optimization.**

Finally, we deeply appreciate jNs7’s rigorous attitude, which motivated us to consider and verify the influence of nonlinear factors in our method (**linearity being a phenomenon, with nonlinearities actually incorporated**).

---
Best regards,

The Authors

---

> ### Author Response · Authors · 2025-12-02
> **Supplement to the revision points of the paper**
>
> ### **Major Revision**:
>
> In Section 3.2.1 ("Outlier Pattern"), we have  revised the narrative by adding detailed illustrations（like Figure 4）. These enhancements clarify our analysis and make the description of outlier patterns more intuitive and accessible to readers.
>
> ### **Minor Revision:**
>
> We have also incorporated a discussion of related work as suggested by the reviewers, focusing on those contributions that are relevant and with which we agree.

---

### Meta-Review · Area_Chair_Ph5i · 2025-12-22

**Summary:**

In this paper, the authors propose a non-uniform quantization method (NuBitQ) to improve the adaptability and efficiency of LLMs. They identify a unique distribution of outliers in non-uniform quantization and, based on this observation, design a plug-and-play OCP module, which is subsequently validated through experiments. The paper received an initial score of 8/6/4/4/4. Most reviewers’ concerns focus on practical efficiency, comparisons with baseline methods, and anomalous experimental results.

In the rebuttal, the authors respond to the reviewers’ comments point by point and largely address these concerns. In particular, for the three reviewers who held negative opinions, the authors addressed their concerns through additional experiments and explanations and further supplemented detailed experiments and clarifications based on the reviewers’ suggestions. This largely alleviates my concerns regarding the issues raised. Nevertheless, providing more comprehensive theoretical analysis or additional experiments would further strengthen the paper’s persuasiveness. At the same time, considering that the proposed method may not demonstrate particularly striking adaptability or efficiency in real-world deployment, this limits its potential to receive higher scores. Overall, I believe that the strengths of this paper outweigh its weaknesses and that the proposed non-uniform quantization method represents a meaningful contribution to existing approaches.

After carefully reviewing the original paper, the reviewers’ comments, and the authors’ responses, I believe that if the reviewers fully engage in the discussion, one or two of the initially negative reviewers may increase their scores, potentially bringing the paper above the borderline. Given that, I recommend a poster acceptance.

**Reviewer Concerns:**

1. Regarding Weakness 1 raised by Reviewer 7kTW, namely that it limits the immediate reproducibility of the method in large-scale deployments, the authors’ response essentially addresses this concern. With respect to system constraints, the authors alleviate the concern to some extent by providing additional experimental results. The supplementary evaluation of different combinations represents a meaningful incremental contribution. Moreover, the reviewer’s question regarding the evaluation of combinations has been addressed, although the experimental details provided by the authors remain insufficiently comprehensive to some degree.
2. The authors have compared their method with the latest competing methods suggested by Reviewer 343g and revised the corresponding descriptions in the main text. Regarding the additional inference overhead raised by multiple reviewers, the authors provide an explanation; however, a more detailed theoretical analysis would be preferable.
3. The issue of anomalous experimental results pointed out by Reviewer R323 has been addressed; however, the authors’ analysis of this phenomenon is not comprehensive. In contrast, the concerns regarding the threshold selection strategy have been adequately addressed.
4. The concerns raised by Reviewer jNs7 have been partially addressed. I agree with the authors’ point that this acceleration method remains meaningful even without reliance on vLLM. In addition, regarding the compatibility between OCP and uniform quantization, the authors could further clarify that the proposed approach is specific to non-uniform quantization. The authors may also consider improving the figures in the revised version to enhance clarity and readability.
5. Reviewer pUW6’s comments are generally positive, and the authors have largely addressed these issues through additional experiments.

**Reviewer Scores:**

In light of the reviewer concerns discussed above, most of the questions raised by Reviewers 343g and 7kTW have been addressed at a fundamental level. It is therefore reasonable to expect that, if they fully engage in the discussion, their scores could be increased to 6.

The original scores from Reviewers pUW6 and R323 were relatively positive and are thus likely to remain unchanged with the rebuttal. In contrast, the concerns raised by Reviewer jNs7 have not been fully addressed, and the score is therefore also likely to remain unchanged.

---

### Decision · Program_Chairs · 2026-01-26

Accept (Poster)